# Previous immunity shapes immune responses to SARS-CoV-2 booster vaccination and Omicron breakthrough infection risk

The heterogeneity of the SARS-CoV-2 immune responses has become considerably more complex over time and diverse immune imprinting is observed in vaccinated individuals. Despite vaccination, following the emergence of the Omicron variant, some individuals appear more susceptible to primary infections and reinfections than others, underscoring the need to elucidate how immune responses are influenced by previous infections and vaccination. IgG, IgA, neutralizing antibodies and T-cell immune responses in 1,325 individuals (955 of which were infection-naive) were investigated before and after three doses of the BNT162b2 vaccine, examining their relation to breakthrough infections and immune imprinting in the context of Omicron. Our study shows that both humoral and cellular responses following vaccination were generally higher after SARS-CoV-2 infection compared to infection-naive. Notably, viral exposure before vaccination was crucial to achieving a robust IgA response. Individuals with lower IgG, IgA, and neutralizing antibody responses postvaccination had a significantly higher risk of reinfection and future Omicron infections. This was not observed for T-cell responses. A primary infection before Omicron and subsequent reinfection with Omicron dampened the humoral and cellular responses compared to a primary Omicron infection, consistent with immune imprinting. These results underscore the significant impact of hybrid immunity for immune responses in general, particularly for IgA responses even after revaccination, and the importance of robust humoral responses in preventing future infections.

Severe acute respiratory syndrome coronavirus 2 (SARS-CoV-2) vaccination has primarily been shown to protect against severe disease but not viral transmission[1]. The high vaccination rates, the higher prevalence of SARS-CoV-2 infections, and the appearance of novel variants have caused a change in the person-to-person transmission dynamics[2]. Because of the novelty of SARS-CoV-2, no or limited immunity existed in the early days of the pandemic. However, this picture has changed due to the continuous SARS-CoV-2 exposure and extensive vaccination campaigns. Despite the efforts, the frequency of breakthrough infections in vaccinated individuals,

e-mail: laura.perez.alos@regionh.dk; peter.garred@regionh.dk

including reinfections in individuals previously infected with SARS-CoV-2 is progressively increasing[2]. This has become particularly relevant after the emergence of the highly contagious Omicron variant and its sublineages. SARS-CoV-2 is continuously evolving and mutations on the spike (S) protein confer profound immune evasion potentials posing a significant threat to antibody therapies and currently authorized Coronavirus Disease 2019 (COVID-19) vaccines[3,4].

It is now well established that both the humoral and cellular immune responses after SARS-CoV-2 vaccination differ highly between individuals. Several contributing factors affecting immunity have been identified, including age, sex, comorbidities, medication, and previous SARS-CoV-2 infections[5–10]. However, there is limited knowledge about the interindividual susceptibility to breakthrough SARS-CoV-2 infections in vaccinated individuals and reinfections, particularly concerning the Omicron variants[11,12]. Despite vaccinations, some individuals seem more prone to primary infections and reinfections than others.

A positive effect of hybrid immunity (a combination of SARS-CoV-2 infection and vaccination) on protecting against SARS-CoV-2 is expected since the commonly used vaccines mount little mucosal IgA responses[13,14]. Concerning systemic IgA immunity, vaccination in infection-naive individuals mounts a weak response, while hybrid immunity mounts a stronger and more sustained IgA response[10]. Whether the systemic IgA response is associated with protection against breakthrough infections is still not resolved.

The concept of immune imprinting, also known as the original antigenic sin, is the immune system's propensity to limit its response to new variant antigens after responding to the original antigen[15]. The consequence of this phenomenon is that the immune system cannot mount more effective responses following new variant infections or vaccines resembling the original immunogen. The phenomenon was first described for the influenza virus and later for human immunodeficiency virus (HIV), dengue fever, and lately, SARS-CoV-2[15–17] and may be relevant for both B- and T-cell immunity.

In this study, we examined the influence of previous infection and vaccination on the degree of reinfection and future infection with Omicron, as well as immune imprinting at the level of antibody isotypes, antibody neutralizing capacity, and T-cell responses. We also explored the potential benefits of hybrid immunity.

## Results

### Characteristics of the study population

The study cohort is part of a longitudinal vaccination study of Danish healthcare professionals described previously[10]. A total of 1325 healthcare professionals were included in the present study, of which 1145 (86.4%) were female, with a median age of 52 (IQR: 41–59) years. All individuals had received a third dose of the BNT162b2 vaccine (boost) at a median of 295 days after administering the first vaccine dose (IQR: 287–302 days, Table 1). At the time of 12 months sample collection, we identified 955 (72.1%) SARS-CoV-2 infection-naive individuals (nucleocapsid [N] protein negative and no positive RT-PCR result). A total of 463 (48.5%) of these had a positive RT-PCR result after the last sampling round, meaning that these individuals were infected after the collection of the 12-month sample and, consequently, after the boost (identified in the text as future infected individuals). We identified 370 individuals with hybrid immunity (a combination of SARS-CoV-2 infection and vaccination). Of these 370, 163 (44.1%) individuals were infected with the Omicron variant (identified in the text as infected with Omicron individuals), whereas 207 individuals were infected with an earlier variant before Omicron dominance in Denmark (identified in the text as infected before Omicron individuals). Of these, 100 were infected with the ancestral variant, 38 with the Alpha variant and 69 with the Delta variant (Table 1). Among participants infected before Omicron, 129 (62.3%) were infected only once, while 78 participants were reinfected by the 12-month sampling or after, confirmed by a positive RT-PCR result (identified in the text as

reinfected individuals). The study design and timeline are illustrated in Fig. 1. Figure 2 depicts a flow chart with the participants and subgroups included in the analyses. The demographic characteristics of the different subgroups are described in detail in Supplementary Table 1. In the reinfection subgroup (identified in the text as reinfected and not reinfected) and future infection subgroup (identified in the text as future infected and not future infected) we evaluated whether certain immune responses are associated with reinfection and breakthrough infections, respectively; and in the immune imprinting cohort (identified in the text as reinfected and infected with Omicron) we evaluated the influence of previous infection on the boost responses.

### IgG levels dynamics after the booster diverge according to infection status, age group, and sex

We fitted a generalized linear-mixed model (GLMM) with five natural cubic splines (NCS) on the 12-month period to study the IgG dynamics over time (Fig. 3). Since the boost administration coincided with the gap between the 6- and 12-month rounds, the insufficient observed data did not allow us to model the immunologic event immediately after vaccination (Fig. 3, gray-shaded area). To show the expected immediate response after the boost, we fitted an additional two-part independent model. One GLMM with two NCS from baseline to 6 months and a linear-mixed model on the 12-month round only. Both models were used to theoretically project the antibody waning until boosting and project the peak reached after boosting (Supplementary Fig. 1, gray-shaded area). Since the GLMM fitted on the entire 12-month period allowed us to evaluate the influence of the initial immune response on the boost effect, statistical analyses were performed using this model.

Due to the large number of predictive values (model outcomes) provided by the different GLMMs, predictive values are reported only on females due to simplicity and power rationale. All predicted values for all age groups, infection status, and sex can be found in the respective tables (Supplementary Tables 2–17) in the Supplementary Information. The time points chosen for direct comparison between groups or time periods were selected based on when the peak level was reached, depending on the age group, infection status, and sex.

IgG dynamics over time were characterized by an initial peak in IgG levels after the second dose (prime), followed by a rapid waning. Administration of a boost resulted in the IgG levels being restored to similar levels as observed following the prime in all three groups defined as (i) infection-naive individuals, (ii) individuals infected before Omicron and iii) individuals infected with Omicron (Fig. 3, left, middle and right panel, respectively). Significant differences in IgG dynamics over time were observed between the three groups (p < 0.001, Fig. 3). Individuals infected before Omicron demonstrated a consistently higher IgG response compared to infection-naive individuals regardless of the age group (e.g., 23,821 Arbitrary Units [AU]/ml [95% confidence interval (CI): 19,433–29,241 AU/ml] in females infected before Omicron aged <40 years compared to 14,999 AU/ml [95% CI: 13,163–17,131 AU/ml] in infection-naive females aged <40 years, after the boost; Supplementary Table 2). Individuals infected before Omicron aged >60 years presented higher IgG levels after the third dose compared to the younger age groups (e.g., 23,821 AU/ml [95% CI: 19,433–29,241 AU/ml] in females aged <40 years, 21,710 AU/ml [95% CI: 18,343–25,733 AU/ml] in females aged 40–60 years, and 41,708 AU/ml [95% CI: 30,683–57,099 AU/ml] females aged >60 years, Supplementary Table 2). As expected, individuals infected with Omicron showed the highest IgG levels following the boost due to the proximity of the last infection to the sampling.

A significant interaction between days from the first vaccine dose and sex was observed (p = 0.011). IgG levels after the boost in infection-naive males were higher than in infection-naive females, contrary to the IgG peak levels observed after the prime. This interaction was only observed in infection-naive individuals but not in individuals with

**Table 1 | Demographic data and characteristics of the main study cohort at the 12-month collection round**

| | Total (N = 1325)[a] | Infection-naive (N = 955)[b] | Infected before Omicron (N = 207)[c] | Infected with Omicron (N = 163)[d] | P-value |
|---|---|---|---|---|---|
| **Sex** | | | | | |
| Female | 1145 (86.4%) | 833 (87.2%) | 180 (87.0%) | 132 (81.0%) | 0.0961[e] |
| Male | 180 (13.6%) | 122 (12.8%) | 27 (13.0%) | 31 (19.0%) | |
| **Age (years)** | | | | | |
| Median (IQR) | 52 (41–59) | 54 (43–61) | 47 (36–57) | 46 (39–55) | <0.0001[e] |
| <40 | 316 (23.8%) | 193 (20.2%) | 70 (33.8%) | 53 (32.5%) | <0.0001[f] |
| >40–60 | 718 (54.2%) | 518 (54.2%) | 107 (51.7%) | 93 (57.1%) | |
| >60 | 291 (22.0%) | 244 (25.5%) | 30 (14.5%) | 17 (10.4%) | |
| **BMI** | | | | | |
| Median (IQR) | 24 (22–27)[g] | 24 (22–27)[h] | 24 (22–27)[i] | 24 (22–26)[j] | 0.3646[e] |
| Underweight | 17 (1.4%) | 10 (1.2%) | 6 (3.1%) | 1 (0.8%) | |
| Normal | 681 (57.9%) | 480 (56.3%) | 115 (60.2%) | 86 (64.7%) | |
| Overweight | 305 (25.9%) | 227 (26.6%) | 43 (22.5%) | 35 (26.3%) | 0.0626[f] |
| Obese | 174 (14.8%) | 136 (15.9%) | 27 (14.1%) | 11 (8.3%) | |
| **Infection status** | | | | | |
| Infected before Omicron | 129 (9.7%) | N.A. | 129 (62.3%) | N.A. | |
| Ancestral variant[k] | 54 (20.2%) | N.A. | 54 (41.9%) | N.A. | |
| Alpha variant | 19 (7.1%) | N.A. | 19 (14.7%) | N.A. | |
| Delta variant | 56 (21.0%) | N.A. | 56 (43.4%) | N.A. | |
| Infected before Omicron and reinfected | 78 (5.9%) | N.A. | 78 (37.7%) | N.A. | |
| Ancestral variant[k] | 46 (28.5%) | N.A. | 46 (59.0%) | N.A. | |
| Alpha variant | 19 (11.8%) | N.A. | 19 (24.4%) | N.A. | N.A. |
| Delta variant | 13 (8.1%) | N.A. | 13 (16.7%) | N.A. | |
| Infected with Omicron | 163 (12.3%) | N.A. | N.A. | 163 (100%) | |
| Infection-naive | 492 (37.1%) | 492 (51.5%) | N.A. | N.A. | |
| Infection-naive but will be infected with Omicron | 463 (34.9%) | 463 (48.5%) | N.A. | N.A. | |
| **Time between first and second dose (days)** | | | | | |
| Median (IQR) | 30 (29–33) | 30 (29–33) | 30 (29–33) | 30 (29–33) | 0.7171[e] |
| **Time between first and third dose (days)** | | | | | |
| Median (IQR) | 295 (287–302) | 294 (287–302) | 298 (291–309) | 294 (285–302) | <0.0001[e] |

*IQR* Interquartile range, *N.A.* Not Applicable.

[a]Participants who contributed with a sample for the cellular immunity studies: 874 individuals.
[b]Participants who contributed with a sample for the cellular immunity studies: 651 individuals.
[c]Participants who contributed with a sample for the cellular immunity studies: 124 individuals.
[d]Participants who contributed with a sample for the cellular immunity studies: 99 individuals.
[e]Kruskal–Wallis test (two-sided) between participants infected before Omicron, participants infected with Omicron and infection-naive participants.
[f]Chi-squared test (two-sided) between participants infected before Omicron, participants infected with Omicron and infection-naive participants.
[g]Missing values: 148 individuals.
[h]Missing values: 102 individuals.
[i]Missing values: 16 individuals.
[j]Missing values: 30 individuals.
[k]Ancestral variant dominance in Denmark comprised a wide combination of the clades 20E (EU1), 20A (EU2), 20A/S:439K, 20B/S:626S, among others[70].

hybrid immunity, where the IgG levels were higher in females than males both after the prime and boost (Supplementary Table 2). The biggest difference in IgG levels between sexes was observed in individuals infected before Omicron (e.g., 23,891 AU/ml [95% CI: 19,433–29,241 AU/ml] in females aged <40 years old compared to 17,202 AU/ml [95% CI: 12,312–23,959 AU/ml] in males aged <40 years old, Supplementary Table 2). Males infected before Omicron had similar IgG levels after the boost compared to the infection-naive males (e.g., 17,202 AU/ml [95% CI: 12,312–23,959 AU/ml] in males infected before Omicron aged <40 years compared to 16,547 AU/ml [95% CI: 13,413–20,496 AU/ml] in infection-naive males aged <40 years, Supplementary Table 2).

Peak IgG levels reached after the prime in infection-naive individuals were higher than the peak IgG levels observed in individuals infected with Omicron regardless of age (e.g., 26,107 AU/ml [95% CI: 21,799–31,208 AU/ml] in infection-naive females aged <40 years

compared to 19,355 AU/ml [95% CI: 14,566–25,612 AU/ml] in females infected with Omicron aged <40 years, Supplementary Table 2).

**Neutralizing antibody levels are enhanced after booster dose**
GLMMs with NCS were used to study the dynamic changes in neutralizing antibodies (nAbs) over time. Significant differences in the nAbs dynamics were observed in all three groups (p < 0.001, Fig. 4). Administration of the boost substantially increased the nAb levels compared to the peak generated after the prime in all groups (Fig. 4). Individuals with hybrid immunity (particularly those infected most recently with Omicron) presented the highest levels of nAbs compared to infection-naive individuals (Supplementary Table 3). Significant changes in the nAbs trends were observed in the different age groups over time (p < 0.001, Fig. 4), where aging was associated with lower nAbs levels after the prime, while similar nAb levels were observed between the different age groups after the boost (Supplementary

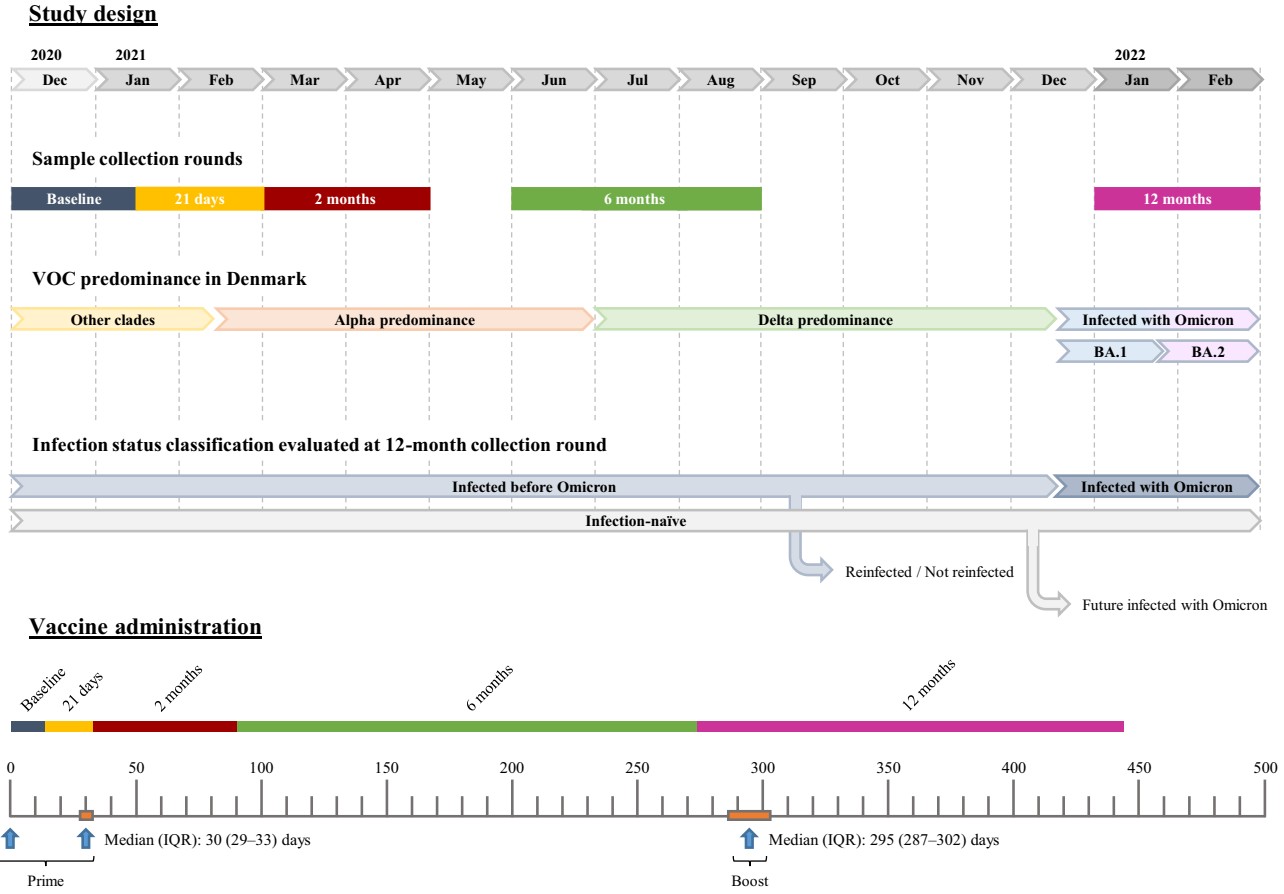

**Fig. 1 | Study design and timeline.** Study design depicting timeline of sample collection, the variants of concern (VOC) dominance in Denmark during the study, and the classification over time of the main cohort according to the infection status of the participants analyzed by the 12-month collection round. Timeline of the administration of the three vaccine doses (vertical blue arrow) in the main cohort and the sample collection rounds. IQR Interquartile range.

Table 3). As observed for IgG, the peak of nAbs following the prime was lower in individuals infected with Omicron compared to infection-naive individuals (e.g. 14,323 International Units [IU]/ml [95% CI: 11,011–18,601 IU/ml] in infection-naive females aged <40 years compared to 11,597 IU/ml [95% CI: 8230–16,402 IU/ml] in females infected with Omicron aged <40 years, Supplementary Table 3). Moreover, an association between nAb levels over time and sex was observed (p = 0.037), where females displayed higher levels of nAb following the prime. However, after the boost, males showed a more substantial increase in nAbs than females. This dynamic change resulted in comparable levels following the booster shot (Supplementary Table 3). A two-part independent model of nAb waning and boost projection are illustrated in Supplementary Fig. 2.

**Hybrid immunity maintains IgA responses**

IgA levels were modeled using GLMMs with a binomial distribution due to the assumptions of non-normally distributed data. An increase in the probability of a positive IgA response after the boost was observed in all groups (Fig. 5). However, the IgA response dynamics differed significantly over time according to the infection status (p < 0.001, Fig. 5). Individuals infected before Omicron maintained a probability of having a positive IgA response above 25% over time, which was enhanced after the boost. Individuals infected more recently (infected with Omicron) showed the greatest increase in the probability of a positive IgA response following boosting and infection (Supplementary Table 4). Infection-naive individuals exhibited a poorer IgA

response after the boost compared with individuals with hybrid immunity (e.g., 33% [95% CI: 22–46%] in infection-naive females aged <40 years compared to 82% [95% CI: 73–90%] in females infected before Omicron aged <40 years, Supplementary Table 4). Individuals who became infected with Omicron in the future had a lower probability of a positive IgA response after the administration of the prime compared with individuals who remained infection-naive (e.g., 67% [95% CI: 57–77%] in infection-naive females compared to 44% [95% CI: 29–59%] in females infected with Omicron aged <40 years, Supplementary Table 4). No significant differences were observed between females and males (p = 0.581, Fig. 5). A two-part independent model of IgA response waning and boost projection are illustrated in Supplementary Fig. 3.

**T-cell-derived IFN-γ levels are boosted after the third dose and correlate with IgG and IgA levels**

Cellular responses were assessed as IFN-γ release from T-cells stimulated with S1 peptides. IFN-γ levels were significantly higher in infection-naive individuals and individuals infected with Omicron after boost administration (12-month sampling) compared to levels before boosting (6-month sampling) (p < 0.001 in both groups, Fig. 6a, c). Individuals infected before Omicron presented higher IFN-γ levels at the 6-month sampling compared with the other groups (p < 0.007, Kruskal–Wallis test). However, no significant difference in IFN-γ levels between the two sampling rounds was observed in this group (p = 0.100, Fig. 6b). Individuals infected with Omicron showed the

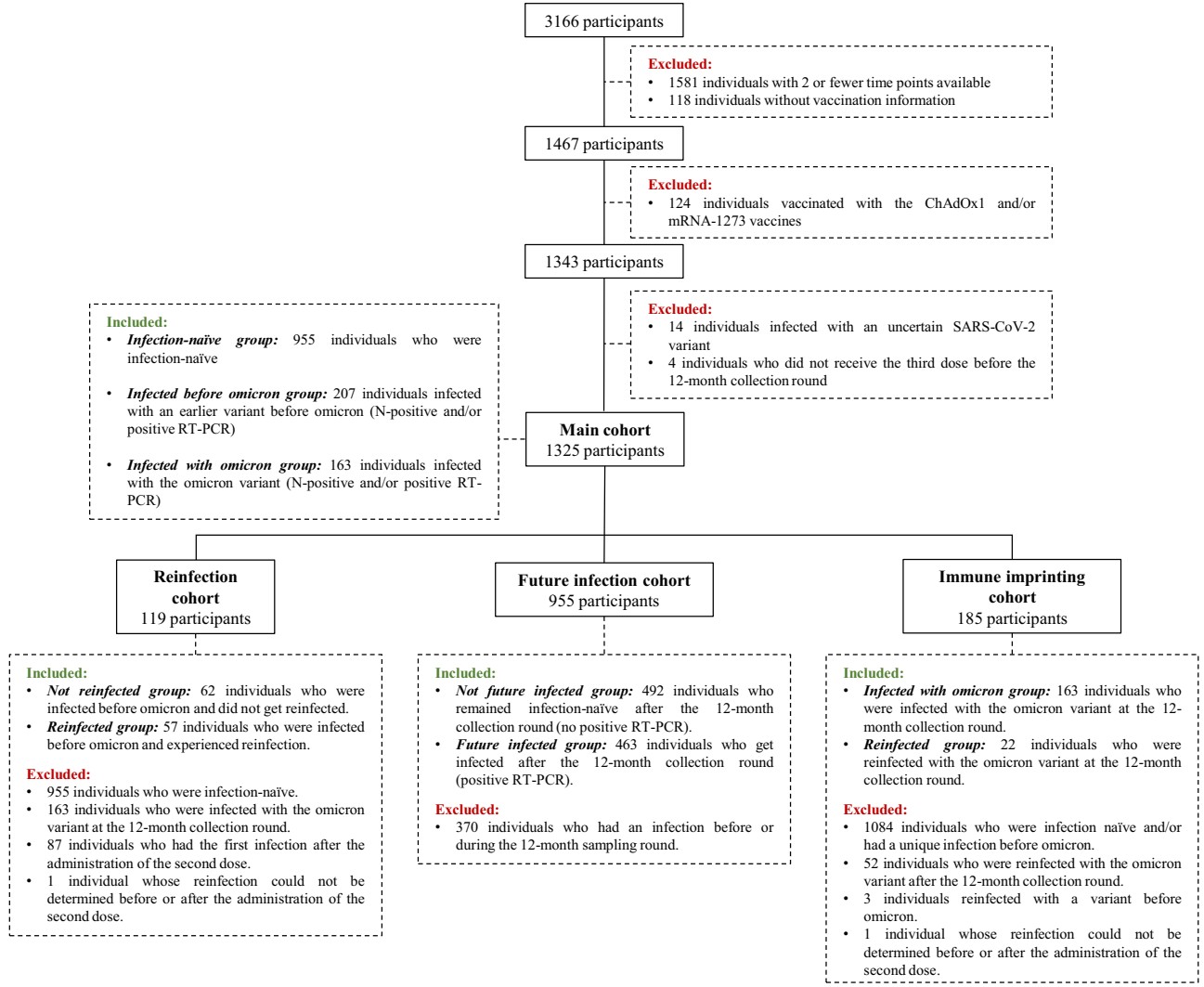

**Fig. 2 | Study flow chart.** Flow chart illustrating the inclusion and exclusion criteria to obtain the main cohort used for statistical analyses and the three subgroups obtained from the main cohort for further statistical studies (reinfection, future infection, and immune imprinting).

highest IFN-γ level response after the boost compared to the other groups (p < 0.001, Kruskal–Wallis test). Comparable results were observed when IFN-γ levels were modeled using a linear-mixed model (Supplementary Fig. 4 and Supplementary Table 5). Significant differences in IFN-γ levels between the different age groups were observed (p = 0.006, Supplementary Fig. 4), characterized by higher levels of IFN-γ in younger individuals (Supplementary Table 5). Males generated lower cellular responses compared to females (p = 0.033), but the dynamics were similar between the sexes (Supplementary Table 5).

Significant correlations between IgG and IFN-γ levels were observed for all groups at both the 6- and 12-month sampling (Fig. 6d, f), being more pronounced at the 12-month sampling after the boost (overall R = 0.47, p < 0.001, Fig. 6f). Correlation between IgA and IFN-γ levels was only evident in individuals infected before Omicron at the 6-month sampling after the prime (R = 0.46, p = 0.026, Fig. 6e). However, at the 12-month sampling, following the boost, a significant correlation between IgA and IFN-γ levels was only observed in infection-naïve individuals (R = 0.21, p < 0.001, Fig. 6g). The overall correlation between IgA and IFN-γ levels was detected at both samplings (6-month sampling: overall R = 0.26, p = 0.001, Fig. 6e; 12-month sampling: overall R = 0.36, p < 0.001, Fig. 6g).

## Decreased humoral responses to the SARS-CoV-2 vaccine are related to reinfections

Differences in IgG, IgA, nAbs, and IFN-γ levels following priming between individuals infected before Omicron who did not experience reinfection (not reinfected) and those who did (reinfected) were studied (Fig. 7). Of note, only individuals who had the first infection before the administration of the second vaccine dose were included to establish reliable comparisons (demographic characteristics in Supplementary Table 1). IgG, IgA, and nAb levels were significantly lower in individuals who experienced reinfection in the future compared to those who did not (p = 0.007, p = 0.024, and p = 0.035, respectively, Fig. 7). There was no significant difference between the groups regarding IFN-γ levels (p = 0.340, Fig. 7). Multiple linear regression analyses showed comparable results (p < 0.001, p = 0.042, and p = 0.035 for IgG, IgA, and nAbs, respectively, Supplementary Table 18). Similar observations were detected using GLMMs. The dynamics of IgG and nAb levels showed significantly different trends between reinfected and not reinfected individuals (p = 0.044 and p = 0.016, respectively; Supplementary Figs. 5 and 6, Supplementary Tables 6 and 7). A tendency was observed regarding the dynamics in the probability of positive IgA responses and IFN-γ levels between reinfected and not reinfected individuals

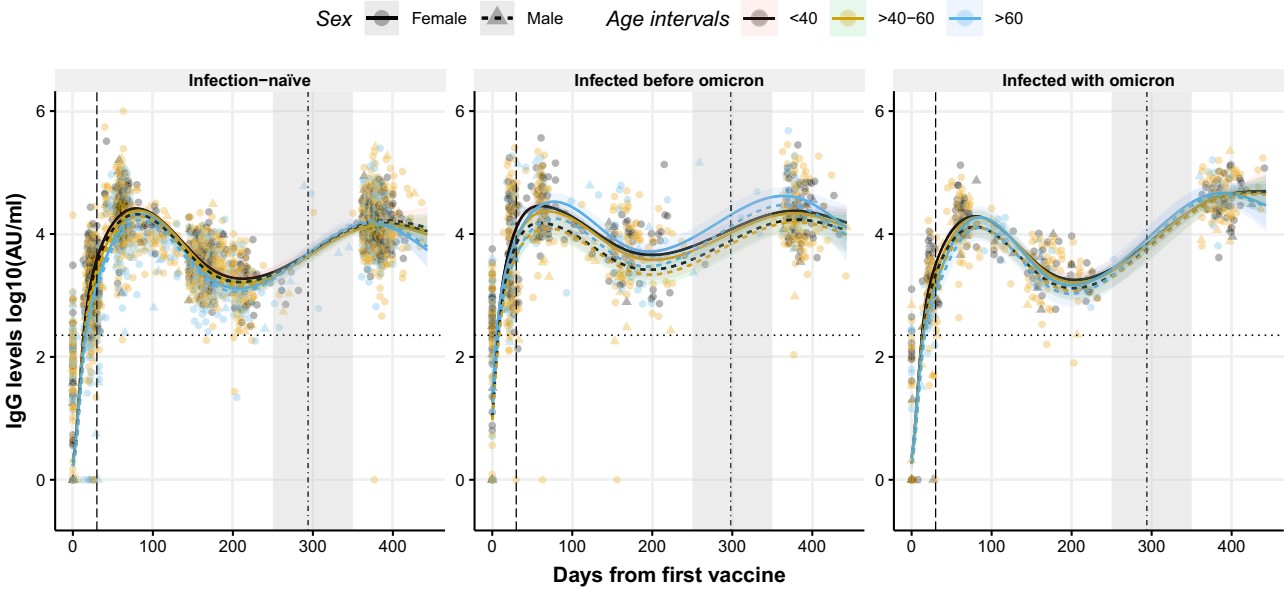

**Fig. 3 | Dynamics of circulating IgG levels against RBD after the first dose of the BNT162b2 vaccine using a non-linear model.** Distribution of IgG levels, represented in log10(AU/ml), over time (days from the first vaccine) in infection-naive individuals (left), in individuals previously infected with a variant before Omicron (middle), and in individuals infected with Omicron (right). Circles and triangles represent the observed levels of circulating IgG levels in females and males, respectively. Solid and dashed lines represent the predicted levels of circulating IgG levels calculated by the model in females and males, respectively. Black, yellow, and blue colors represent individuals with age <40, 40–60, and >60 years, respectively. Horizontal black dotted line represents the threshold for assay positivity. Vertical dashed and dashed-dotted lines represent when the second and the third dose was administered, respectively (median days). Shadowed areas represent the 95% confidence interval. Centre for the confidence interval is the predicted (mean) values. Predictive values in the gray-shaded area (days 250–350) do not represent a realistic projection due to insufficient observed data to provide realistic predictive data. Source data are provided as a Source Data file.

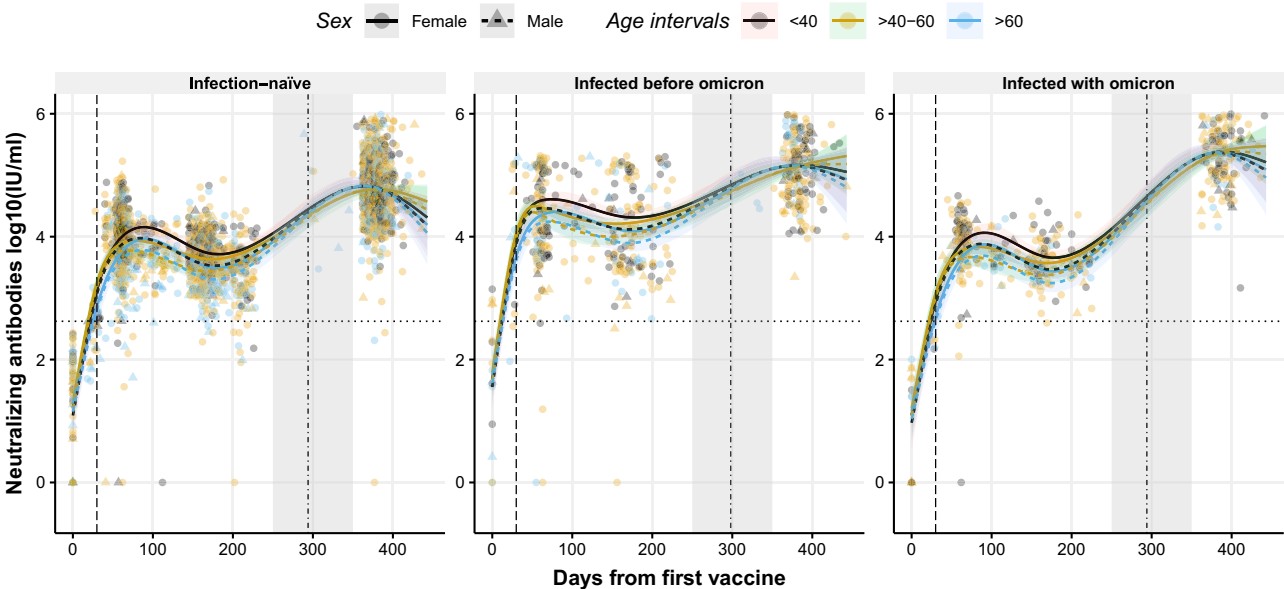

**Fig. 4 | Dynamics of circulating neutralizing antibody levels against RBD after the first dose of the BNT162b2 vaccine using a non-linear model.** Distribution of neutralizing antibody levels, represented in log10(IU/ml), over time (days from the first vaccine) in infection-naive individuals (left), in individuals previously infected with a variant before Omicron (middle), and in individuals infected with Omicron (right). Circles and triangles represent the observed levels of circulating neutralizing antibody in females and males, respectively. Solid and dashed lines represent the predicted levels of circulating neutralizing antibody calculated by the model in females and males, respectively. Black, yellow, and blue colors represent individuals with age <40, 40–60, and >60 years, respectively. Horizontal black dotted line represents the threshold for assay positivity. Vertical dashed and dashed-dotted lines represent when the second and the third dose was administered, respectively (median days). Shadowed areas represent the 95% confidence interval. Centre for the confidence interval is the predicted (mean) values. Predictive values in the gray-shaded area (days 250–350) do not represent a realistic projection due to insufficient observed data to provide realistic predictive data. Source data are provided as a Source Data file.

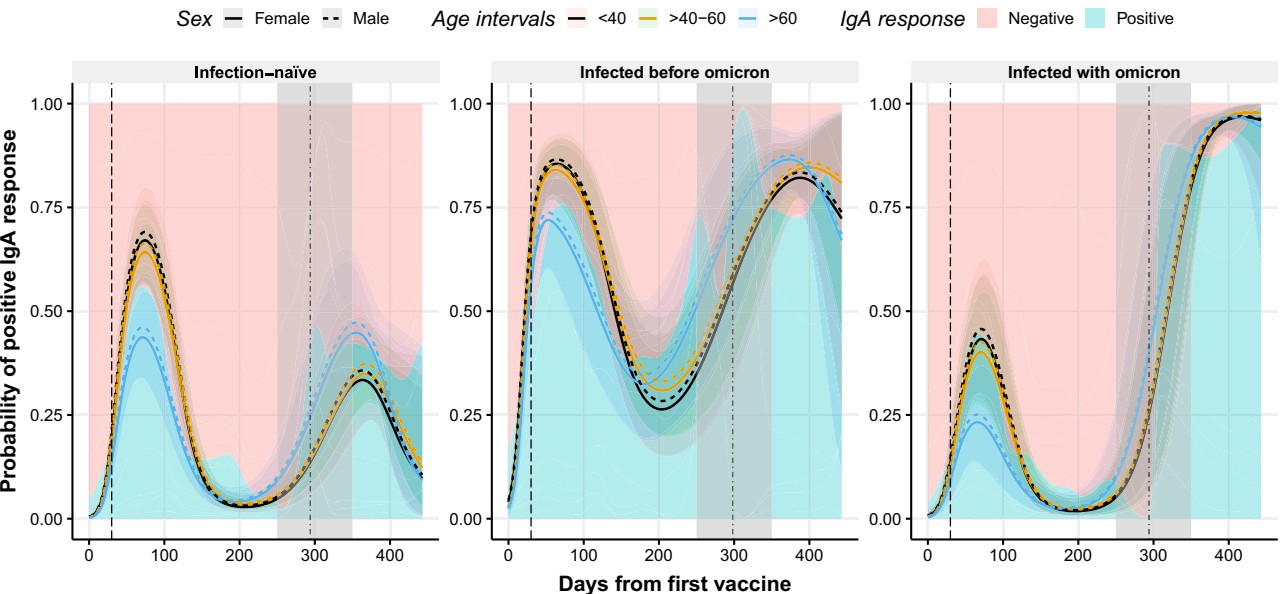

**Fig. 5 | Observed and predicted probability of positive IgA responses against RBD after the first dose of the BNT162b2 vaccine.** Distribution of positive IgA response (probability) over time (days from the first vaccine) in infection-naïve individuals (left), in individuals previously infected with a variant before Omicron (middle), and in individuals infected with Omicron (right). Blue and pink backgrounds represent the conditional density estimation of positive and negative IgA responses, respectively. Solid and dashed lines represent the predicted probability of positive IgA responses calculated by the model in females and males, respectively. Black, yellow, and blue colors represent individuals with age <40, 40–60, and >60 years, respectively. Vertical dashed and dashed-dotted lines represent when the second and the third dose was administered, respectively (median days). Shadowed areas represent the 95% confidence interval. Centre for the confidence interval is the predicted (mean) values. Predictive values in the gray-shaded area (days 250–350) do not represent a realistic projection due to insufficient observed data to provide realistic predictive data. Source data are provided as a Source Data file.

(p = 0.099 and p = 0.118, respectively; Supplementary Figs. 7 and 8, Supplementary Tables 8 and 9).

### Lower humoral responses after the third dose are associated with future infections

To evaluate the association between immune responses and the risk of future infections, IgG, IgA, nAb, and IFN-γ levels were studied in a subcohort of infection-naive individuals who remained uninfected (not future infected) after the 12-month sampling and infection-naive individuals who became infected after sample collection assessed by a positive RT-PCR test (future infected) (Fig. 8, demographic characteristics in Supplementary Table 1). The IgG, IgA and nAb levels were significantly lower in individuals who would experience a future infection than those who did not after the 12-month collection round (p = 0.009, p = 0.031, and p = 0.028 for IgG, IgA and nAb levels, respectively, Fig. 8). No significant difference in IFN-γ levels was observed between the groups (p = 0.510, Fig. 8). Multiple linear regression analyses showed comparable results (p = 0.018, p = 0.040, and p = 0.902 for IgG, IgA, and IFN-γ, respectively, Supplementary Table 18), although a tendency was observed for nAbs (p = 0.103). When using GLMMs, similar trends were detected, where the IgG and nAb levels and probability of positive IgA responses dynamics over time differed significantly between individuals with a future infection and those who were not future infected (p = 0.023, p = 0.028, and p = 0.028 for IgG, nAbs and IgA, respectively, Supplementary Figs. 9–11, respectively, Supplementary Table 10, 11 and 12). Significant differences in the IFN-γ levels dynamics were also observed (p = 0.006, Supplementary Fig. 12, Supplementary Table 13).

### Humoral and cellular vaccine responses are influenced by previous immune imprinting

To assess the impact of possible imprinting from previous SARS-CoV-2 variants on the vaccine immune response, we studied IgG, IgA, nAb, and IFN-γ levels in a subcohort of individuals exposed to an earlier SARS-CoV-2 variant who experienced reinfection with Omicron (reinfected) and infection-naive individuals who experienced a primary infection with Omicron (infected with Omicron). In both groups, infections occurred before the sample collection following the boost (demographic characteristics in Supplementary Table 1). IgG, IgA and nAb levels were significantly increased after the boost in individuals infected with Omicron compared to those who were reinfected (p < 0.001, p = 0.013, and p = 0.019 in IgG, IgA, and nAb levels, respectively, Fig. 9). There was no significant difference between groups regarding IFN-γ levels (p = 0.270, Fig. 9). Multiple linear regression analyses showed comparable results (p < 0.001, p = 0.036, p = 0.055, and p = 0.679 for IgG, IgA, nAbs, and IFN-γ, respectively, Supplementary Table 18). When modeling using GLMMs, significantly different dynamics over time were detected for IgG and nAbs (p < 0.001 for both, Supplementary Figs. 13 and 14, respectively), characterized by a greater increase in IgG and nAb levels in individuals infected with Omicron compared to reinfected individuals following boosting (Supplementary Tables 14 and 15, respectively). Borderline significant differences in the dynamics over time of the probability of positive IgA responses between groups were observed (p = 0.064, Supplementary Fig. 15). Individuals infected with Omicron were lacking a positive IgA response before boost administration and subsequent Omicron infection. Consequently, these individuals showed a marked increase in IgA response compared to reinfected individuals (Supplementary Table 16). IFN-γ level dynamics differed significantly over time (p = 0.001, Supplementary Fig. 16, Supplementary Table 17), with a clear increase in IFN-γ levels in individuals infected with Omicron.

## Discussion

The SARS-CoV-2 vaccination campaign has demonstrated a significant reduction in the severity of COVID-19 disease, thereby decreasing the number of hospital admissions and mortality rates[18]. Despite this achievement, waning immunity following vaccination and the immune

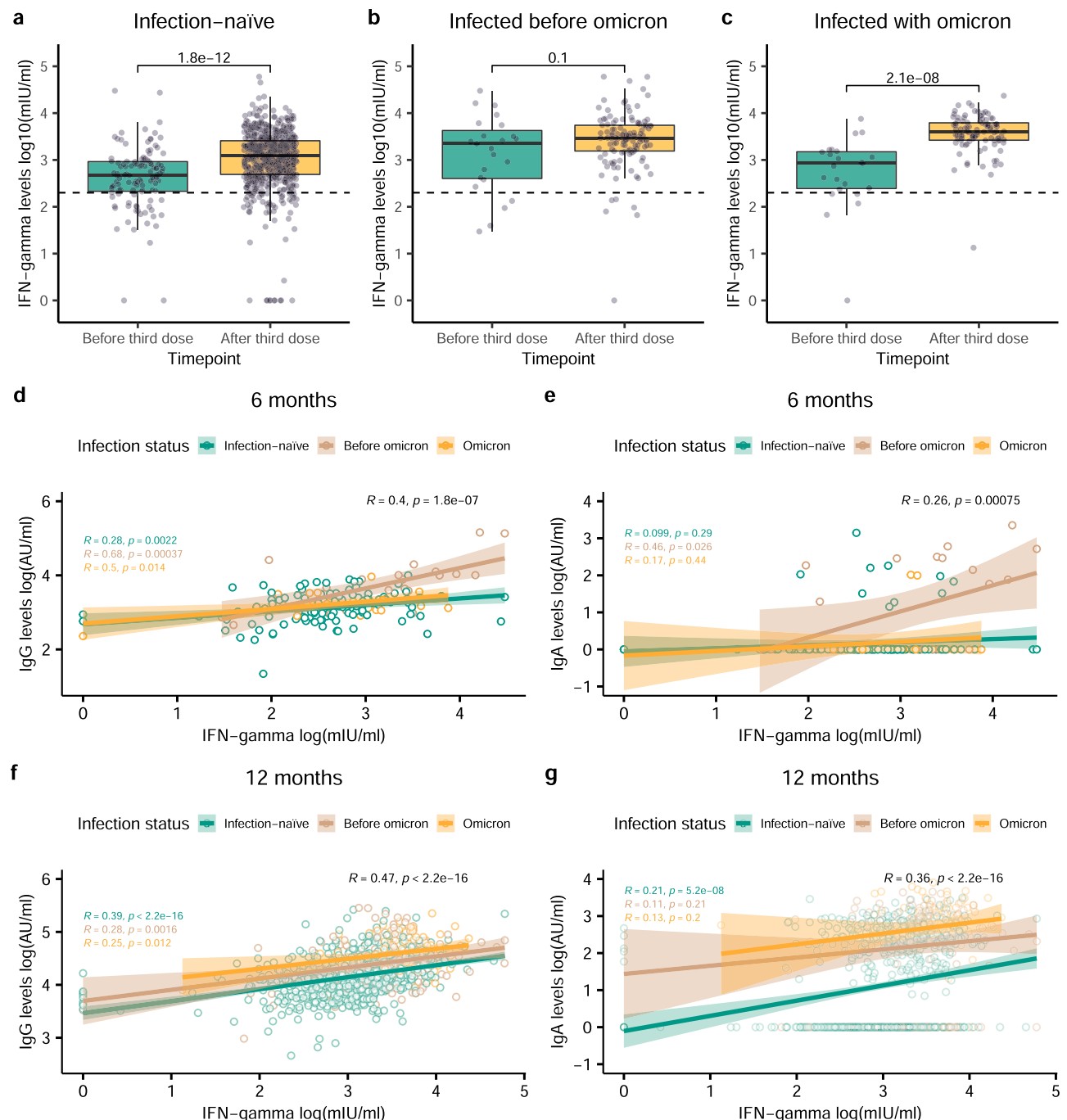

**Fig. 6 | Distribution of IFN-γ released from activated T-cells and correlation with antibody responses.** **a–c** IFN-γ levels collected before (6-month round; green) or after (12-month round; yellow) of the third dose administration, represented in log10(mIU/ml), in infection-naive individuals (n = 115 and n = 650 biologically independent samples before and after the third dose administration) (**a**), individuals previously infected with a variant before Omicron (n = 23 and n = 122 biologically independent samples before and after the third dose administration) (**b**), and in individuals infected with Omicron (n = 23 and n = 98 biologically independent samples before and after the third dose administration) (**c**). Circles represent observed data. Data reported as median and interquartile range (box),

whiskers represent 1.5 times the interquartile range. Dashed horizontal line indicates assay positivity threshold. P-values were calculated using Mann-Whitney U test (two-sided). Correlation between IFN-γ levels with IgG (**d**) and IgA levels (**e**) at 6-month round. Correlation between IFN-γ levels with IgG (**f**) and IgA levels (**g**) at 12-month round. Blue, red, yellow colors represent infection-naive individuals, individuals infected before Omicron, and individuals infected with Omicron, respectively. Black color represents overall Spearman Rank test results. Circles represent observed data. Shadowed areas represent the 95% confidence interval. P-values were calculated using Spearman Rank test (two-sided). p < 0.05 was considered statistically significant. Source data are provided as a Source Data file.

evasion challenge introduced by the emergence of novel SARS-CoV-2 variants, such as Delta and Omicron, has resulted in the need for vaccine boosters[19,20]. Immune responses after BNT162b2 boosting have been investigated; however, limitations including small study

cohorts, lack of IgA data, time from vaccination being addressed as a categorical variable, or exclusion of individuals with different infection statuses restrict the findings[21–24]. Here, we provide a comprehensive study of the effect of the BNT162b2 COVID-19 vaccine and infection as

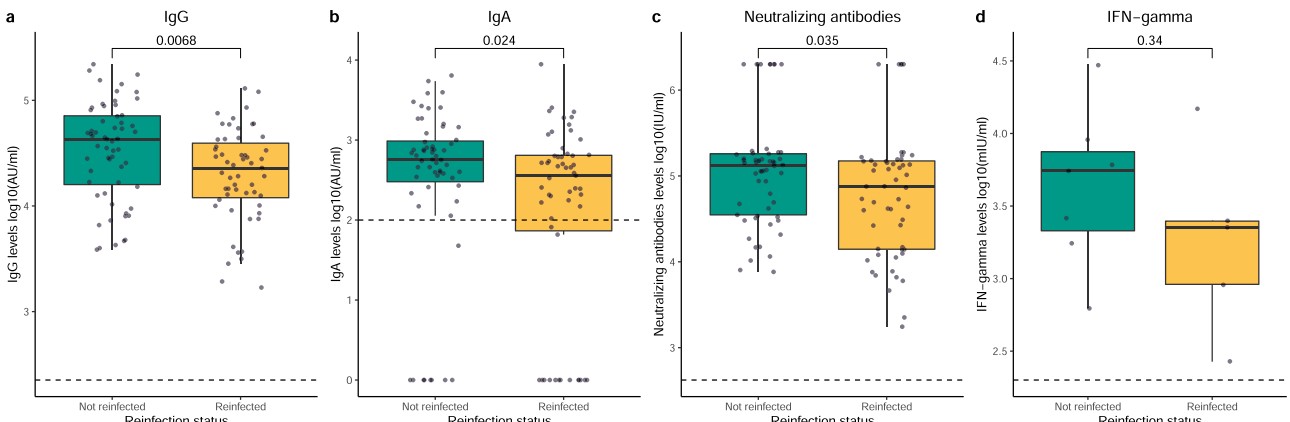

**Fig. 7 | Humoral and cellular responses at the waning period in individuals previously infected before Omicron in relation to reinfection.** Distribution of IgG levels (**a**) and IgA levels (**b**), both represented as log10(AU/ml) (n = 59 and n = 55 biologically independent samples in the not reinfected and reinfected groups, respectively); neutralizing antibody levels (**c**), represented as log10(IU/ml) (n = 57 and n = 55 biologically independent samples in the not reinfected and reinfected groups, respectively); and IFN-γ levels (**d**), represented in log10(mIU/ml) (n = 7 and n = 5 biologically independent samples in the not reinfected and reinfected groups, respectively); during the waning period (day 15 after the second dose and day -1 before the third dose) in individuals infected before Omicron who did not get a second SARS-CoV-2 infection (Not reinfected) and those who did get a second SARS-CoV-2 infection (Reinfected). Green and yellow colors represent not reinfected and reinfected individuals, respectively. Circles represent observed data. Data reported as the median and interquartile range (box), whiskers represent 1.5 times the interquartile range. Dashed horizontal line indicates the threshold for assay positivity. P-values were calculated using Mann–Whitney U test (two-sided), where p < 0.05 was considered significant. Source data are provided as a Source Data file.

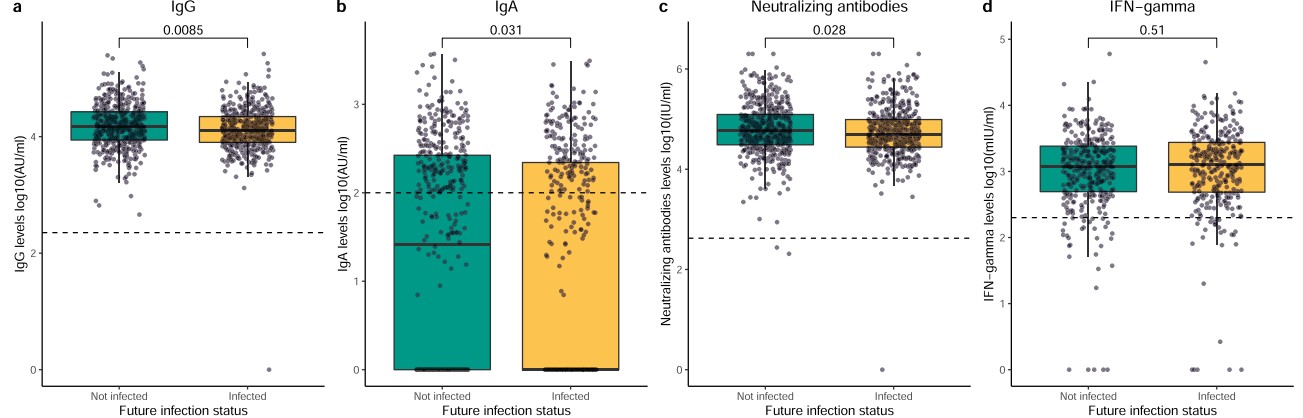

**Fig. 8 | Humoral and cellular responses after the third dose in infection-naive individuals in relation to future infection.** Distribution of IgG levels (**a**) and IgA levels (**b**), both represented as log10(AU/ml) (n = 492 and n = 460 biologically independent samples in the not infected and infected groups, respectively); neutralizing antibody levels (**c**), represented as log10(IU/ml) (n = 488 and n = 459 biologically independent samples in the not infected and infected groups, respectively); and IFN-γ levels (**d**), represented in log10(mIU/ml) (n = 340 and n = 308 biologically independent samples in the not infected and infected groups, respectively); generated after the third dose (day 15 after the third dose) in infection-naive individuals who remain infection-naive after the 12-month round (Not infected) and those who get a future Omicron infection after the 12-month round based on a positive RT-PCR result (Infected). Green and yellow colors represent not infected and infected individuals, respectively. Circles represent observed data. Data reported as the median and interquartile range (box), whiskers represent 1.5 times the interquartile range. Dashed horizontal line indicates the threshold for assay positivity. P-values were calculated using Mann–Whitney U test (two-sided), where p < 0.05 was considered significant. Source data are provided as a Source Data file.

a model on the humoral and cellular dynamics in a large population of apparently healthy individuals with diverse imprinted immunity against the SARS-CoV-2 virus.

We identified diverse IgG level dynamics after the third COVID-19 vaccine dose. Although a general increase in immune responses against the SARS-CoV-2 receptor-binding domain (RBD) was detected, the infection status, as well as the age group and sex, had an impact on the magnitude of the response. It has been reported that neither age nor sex is an influencing factor on IgG levels after the booster[25-27]. Here, we observe that sex is still an influencing factor, specifically in individuals who have had an infection with a variant before Omicron. Interestingly, neither age nor sex influenced the nAbs levels following booster vaccination, illustrating a discrepancy between factors influencing IgG titers and neutralizing capacity. This could be attributed to

antibody affinity maturation which occurs over time and particularly after the booster dose, as observed by others[28], being more evident in individuals with hybrid immunity.

Despite the humoral and cellular increase after the booster dose, it has been reported that vaccine effectiveness wanes rapidly compared to the prime doses, probably due to the high incidence of the Omicron variant[24], leading to breakthrough infections and reinfections. In the present study, we observed a higher reinfection rate among individuals infected before Omicron (37.5%) compared to other studies[29,30]. We also observed that nearly half of the infection-naive individuals (48.4%) became infected with SARS-CoV-2 (Omicron) for the first time following the booster dose. The high infection rates are probably related to the dominance of the Omicron variant in Denmark during the sampling period and because of the discontinuation of

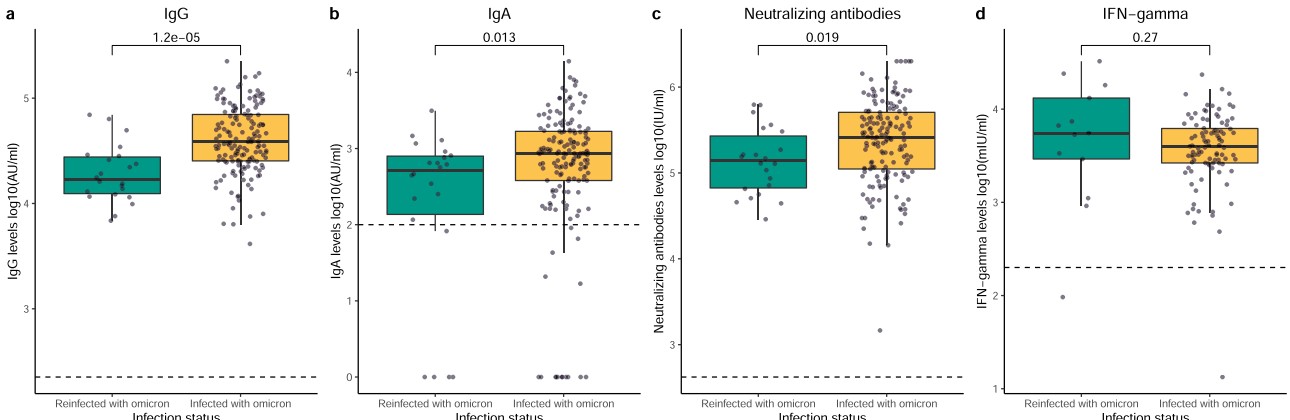

**Fig. 9 | Humoral and cellular responses after the third dose in previously infected individuals in relation to immune imprinting status.** Distribution of IgG levels (**a**) and IgA levels (**b**), both represented as log10(AU/ml) (n = 22 and n = 163 biologically independent samples in the reinfected with Omicron and infected with Omicron groups, respectively); neutralizing antibody levels (**c**), represented as log10(IU/ml) (n = 22 and n = 161 biologically independent samples in the reinfected with Omicron and infected with Omicron groups, respectively); and IFN-γ levels (**d**), represented in log10(mIU/ml) (n = 13 and n = 98 biologically independent samples in the reinfected with Omicron and infected with Omicron groups, respectively); generated after the third dose (day 15 after the third dose) in individuals infected before Omicron who were reinfected with Omicron (Reinfected with Omicron) and individuals infected with Omicron for the first time (Infected with Omicron) at the 12-month round. Green and yellow colors represent reinfected with Omicron and infected with Omicron, respectively. Circles represent observed data. Data reported as the median and interquartile range (box), whiskers represent 1.5 times the interquartile range. Dashed horizontal line indicates the threshold for assay positivity. P-values were calculated using Mann-Whitney U test (two-sided), where p < 0.05 was considered significant. Source data are provided as a Source Data file.

COVID-19 restrictions by the end of January 2022, leading to a rise in viral transmission rates.

Breakthrough infections are a challenge of the currently approved vaccines by the EMA (European Medicine Agency), which requires a detailed evaluation of the mechanism underneath. Therefore, we assessed whether certain immune responses to the BNT162b2 vaccine could be associated with breakthrough infections or reinfections. Individuals infected with the Omicron variant by the 12-month sampling had significantly lower levels of IgG, IgA and nAbs following the prime compared to infection-naive individuals, the major difference being the IgA levels. Whether these individuals infected with Omicron by the 12-month sampling had exposure to seasonal human coronaviruses prior to the first vaccination is unknown. However, cross-reactive responses between different coronaviruses have been reported[31–35] and back-boosting might occur, suggesting that previous immune imprinting in vaccine-naive individuals might modulate the immune system at the time of vaccine priming. Nevertheless, this notion should be considered hypothetical, requiring in-depth B memory cell studies and granular data to elucidate further.

Similar trends were observed in individuals who would become infected with Omicron after the sample collection as they had lower levels of IgG, IgA and nAbs after the boost compared to those who remained infection-naive in the same period. No significant differences between vaccination intervals were observed between future infected individuals and those who remained infection-naive, neither in relation to sex nor age. The findings suggest that interindividual differences in the speed of waning immunity increases the risk of breakthrough infection with the Omicron variant.

Several studies have reported greater immune responses in individuals with hybrid immunity compared to infection-naive individuals;[36–38] here, we can substantiate these findings. Hybrid immunity has been shown to confer more robust protection against future SARS-CoV-2 reinfections, probably due to the broader recognition of different antigens not included in the vaccine design and boosting mucosal immunity[37,39]. Nevertheless, reinfections in previously SARS-CoV-2 infected individuals are still common due to the highly mutated Omicron variant and its sublineages. A high proportion of individuals infected with variants before Omicron were reinfected after boost administration. Here, we report a weaker humoral response

in individuals who experience reinfections in the future, characterized by a lower peak generated and/or a more marked waning after the prime compared to individuals who do not experience reinfections.

Taken together, lower humoral responses following vaccination are associated with increased risk of breakthrough infections and reinfections, as reported previously[7,40,41]. This observation, however modest, was particularly evident in relation to circulating IgA levels, which were negative or close to the positivity threshold in individuals who experienced a breakthrough infection or reinfection. The extend of circulating IgA contribution to protection against infection is not clear and discrepancies regarding its association with mucosal IgA have been reported[42–46]. Circulating IgA function has been described in relation to induction of proinflammatory cellular functions, such as phagocytosis, antibody-dependent cellular cytotoxicity (ADCC), degranulation, antigen presentation, and release of inflammatory mediators[47,48]. A substantial serum IgA-related SARS-CoV-2 neutralizing activity has been shown, being more evident in previously infected individuals and enhanced upon vaccination[43]. Furthermore, a protective role of serum IgA against SARS-CoV-2 infection has been described[49,50], although the protection appears short-term and modest compared to IgG[43,50]. The role of circulating IgA in the protection against SARS-CoV-2 remains elusive and further investigations are needed.

Neutralizing activity against Omicron is reduced compared to previous variants[44,51]. However, cellular immunity appears to remain unaltered after vaccination, suggesting it is the main mechanism preventing COVID-19 severity outcomes but not so efficiently protecting against SARS-CoV-2 transmission and thus breakthrough infections[44,51]. This supports the notion that humoral responses are more important for viral transmission and thus breakthrough infections than cellular responses, as IgG antibodies have been correlated with protection against infection[52]. However, the importance of IgA antibodies, regarded as a primary defense mediator on mucosal surfaces, remains unclear in this context, even within SARS-CoV-2 infected individuals. We cannot confidently provide the origin of the IgA measured in circulation or whether infected and naive individuals present similar specific IgA portfolios.

Depending on the previous exposure to SARS-CoV-2, immune responses to vaccination differ between individuals, which could

be attributed to immune imprinting[15–17,53]. To test this hypothesis, we compared the immune responses between individuals previously exposed to SARS-CoV-2 with an earlier variant (e.g., Ancestral, Alpha, Beta, Delta) and subsequently reinfected with Omicron and infection-naive individuals who experienced a primary infection with Omicron. It has been reported that Omicron reinfections limit antibody boosting in individuals previously infected with a distant strain[31,54]. We also observed that humoral responses in this population were less pronounced compared to previously infection-naive individuals with a primary Omicron infection, following the hybrid immune-damping phenomenon described by Reynolds et al.[54]. In contrast, infection-naive individuals exposed to Omicron presented a very different dynamic response, which was significantly boosted[31,54]. These higher levels against ancestral RBD might be related to the recognition of conserved epitopes, driving the proliferation of memory B-cells boosting the original antibody response[55–58]. However, focus on antibody maturation and breadth recognition of other variants are necessary. Moreover, further investigations are required to evaluate whether this phenomenon, traditionally observed in the influenza vaccination strategy, will negatively hamper the immune response against future infections with novel SARS-CoV-2 variants[54].

Cellular immunity, specifically T-cell immunity, is vital to limiting infection by viral clearance and provides clinical protection in COVID-19[59]. The changes in IFN-γ level dynamics after the booster dose were observed to be less pronounced than the humoral dynamics. Nevertheless, a significant increase in IFN-γ levels was observed after the boost in infection-naive individuals and individuals infected with Omicron, but not in individuals infected before Omicron. Thus, immune imprinting may also play a role in T-cell responses.

No difference in IFN-γ levels was observed between vaccinated individuals experiencing reinfection with Omicron compared to vaccinated individuals experiencing a primary Omicron infection, as observed elsewhere[54,60,61]. This shows that vaccination generates a robust cellular response, which can be boosted after the first antigen exposure. We could not distinguish between IFN-γ released from CD4+ and CD8+ T-cells but other studies have reported predominant and stronger responses from CD4+ T-cells upon peptide stimulation[62–64]. Consequently, we hypothesize that the observed correlation between IFN-γ and IgG levels might be linked to the interplay between CD4+ T-cells eliciting effective B-cell responses, as reported elsewhere[62].

In this study, we have investigated the humoral and cellular response dynamics in a comprehensive large-scale fashion. However, some limitations of the study are pertinent to acknowledge. Due to study design and logistic limitations, we did not collect pre-boost samples, which could influence the model fit. Quantification of the humoral and cellular responses was assessed using the ancestral strain RBD and S1 peptides, respectively. Therefore, we cannot exclude an underestimation of responses following infection due to the Omicron variant in individuals infected or reinfected, as variant-specific antigens were not included in the study. Despite this, using the original strain antigens allowed us to directly evaluate the influence of the BNT162b2 vaccine, designed from the ancestral S protein. It should be emphasized that IgA responses could be underestimated due to the sensitivity of the assay. The study cohort was represented mostly by females, the predominant sex in the healthcare sector in Denmark, which might skew the study. However, no significant differences in sex distribution were observed in the study subgroups. We could have underestimated the exact number of future infected individuals due to changes in the testing strategy in Denmark in the spring 2022. Moreover, viral exposure frequencies between infected/reinfected or not infected/reinfected individuals are inherently uncertain and could introduce bias. Additionally, we did not have information regarding chronic diseases or medication use among participants. However,

there are several strengths worth mentioning; from a more epidemiological point of view, the large number of subjects with many repeated samples provided great power when analyzing the data. In addition, the large cohort allowed us to evaluate diverse infection status subgroups. Still, one of the greatest strengths is the use of GLMMs, a useful tool for modeling dynamics when repeated samples are not available at all time points and the possibility to adjust results for different covariates such as age groups and sex[10,23]; although variables such as time from infection should be considered carefully. Nevertheless, it should be mentioned that although these models can show predicted values at any given time point, the time range with limited observed data available (days 250–350 after first dose) illustrated predicted data that appear biologically unrealistic. Thus we avoided assessing predicted values in this time range and only report when observed data demonstrated greater consistency. The two-part independent model allowed us to project the immunological event of the boost.

The heterogeneity of the SARS-CoV-2 immune responses has become considerably more complex over time due to the different vaccine boosters, the number of antigen exposures and the frequent mutations of the virus. This complicates the vaccine strategy in future vaccination campaigns. Evaluation of both humoral and cellular responses, including the establishment of consensus about correlates of protection, is crucial to identify who may be eligible for additional vaccine boosters. Moreover, novel vaccine designs are required to improve IgA responses, which are primarily enhanced upon SARS-CoV-2 infection, suggesting it is a central component in immune protection.

In conclusion, we demonstrate a modulation of the humoral and cellular responses after the booster dose, primarily influenced by the previous immunity of the individual. SARS-CoV-2 infection has an impact on inducing a robust IgA response after vaccination. Low IgG, IgA, and nAbs responses, but not T-cell responses, are associated with an increased risk of future SARS-CoV-2 infections. Finally, primary infection before Omicron and subsequent reinfection with Omicron significantly dampened the humoral and cellular response, consistent with immune imprinting.

## Methods

### Study design and participants

The study cohort was composed of healthcare professionals from Rigshospitalet and Herlev-Gentofte University Hospital (Capital Region of Denmark) who participated at different sample collection time points in the 12-month prospective longitudinal observational period. Results of immune responses from previous rounds have been reported elsewhere[10,65]. Sample collection did not interfere with the Danish COVID-19 vaccination program[10,65]. Participants included in the 12-month cohort were fully vaccinated with BNT162b2 (Comirnaty) COVID-19 vaccine (Pfizer-BioNTech) and received a booster dose from the same vaccine. Sample collection times spanned from baseline, 21 days, 2 months, 6 months, and 12 months approximately after the first dose. Participants filled out questionnaires with information regarding sex, age, height, and weight. Research Electronic Data Capture (REDCap) was employed to collect and manage the data[66]. Individuals included in the study received oral and written information before providing informed consent. Venous blood samples for humoral analyses were collected from baseline up to 443 days after administering the first vaccine dose at a different sampling collection round (baseline, 21 days, 2 months, 6 months, and 12 months approximately after the first vaccine dose). The total number of repeated measurements per participant was between 3 and 5. Venous blood sampling for measurement of T-cell responses was performed only at the 6- and 12-month collection rounds and the number of repeated measurements per participant was from 1 to 2. Venous blood sample collection fulfilled the principles described in the Declaration

of Helsinki. The protocol was approved by the Regional Scientific Ethics Committee of the Capital Region of Denmark (H-20079890).

### Estimation of anti-SARS-CoV-2 antibody levels

Quantitative determination of antibody levels (IgG and IgA) in plasma against the SARS-CoV-2 ancestral spike S protein RBD were measured using a national validated in-house ELISA-based assay, as described before[10]. A detailed description of the assay can be found in the Supplementary Information. Detection of total antibodies against the N protein was performed using the Elecsys® Anti-SARS-CoV-2 assay (Roche Diagnostics) on the COBAS 8000 platform (e801 module) following the manufacturer's instructions. N protein antibody detection was used as a proxy to determine previous SARS-CoV-2 infections.

### ACE-2/RBD antibody inhibition quantification

Quantitative estimation of the inhibition degree of virus-neutralizing antibodies against RBD to bind the ACE-2 host receptor was performed using a surrogate in-house ELISA-based pseudo-neutralizing as previously described[67]. The pseudo-neutralizing assay used in this study has a high correlation ($r = 0.9231$) with the gold standard plaque reduction neutralization test[67]. A detailed description of the assay can be found in the Supplementary Information.

### T-cell stimulation and IFN-γ quantification

Stimulation of T-cells in fresh whole blood samples with peptides derived from the S1 subunit of S protein was performed using the SARS-CoV-2 IGRA stimulation tube set (ET 2606-3003, EUROIMMUN) following manufacturer's instructions; specific incubation time is described previously[10]. Quantitative IFN-γ released from stimulated T-cells was determined using an IFN-γ ELISA kit (ET 6841-9601, EUROIMMUN) following the manufacturer's instructions. A detailed description of the assay can be found in the Supplementary Information.

### Data interpolation and definitions

Interpolation of circulating IgA and IgG levels, neutralizing antibody levels and IFN-γ levels was executed utilizing GraphPad Prism version 9.3.1 (GraphPad Software) using non-linear regression with four-parameter curve fitting. Interpolated levels of IgA and IgG antibodies were given in AU/ml, being the highest concentration of the calibrator at 200 AU/ml. The threshold for assay positivity was set to 100 and 225 AU/ml for IgA and IgG, respectively. Interpolated neutralizing antibody levels were given in IU/ml, the highest concentration of the calibrator 520 IU/ml (the calibrator was previously quantified into IU/mL using The Working Reagent for anti-SARS-CoV-2 immunoglobulin 21/234, NIBSC). The threshold for assay positivity was set to 420 IU/ml. Interpolated IFN-γ levels were given in mIU/ml. The threshold for assay positivity was set to 200 mIU/ml.

SARS-CoV-2 infection was defined as an individual with detectable antibodies against protein N and/or a positive RT-PCR result. RT-PCR data was collected from the Danish Microbiology Database (MiBa)[68]. SARS-CoV-2 reinfection was defined as an individual with two consecutive positive RT-PCR results separated by 60 or more days[29,69]. Individuals lacking a first RT-PCR confirmation in SARS-CoV-2 infections detected in 2020 (protein N positive) were defined as reinfected when a positive RT-PCR result was performed 60 days or more after the first protein N seropositive sample. Hybrid immunity was defined as an individual being vaccinated with a COVID-19 vaccine and being infected with the SARS-CoV-2 virus. SARS-CoV-2 infections with the Omicron variant were defined by an individual with a positive RT-PCR result obtained from a test after the 20th of December 2021, when the Omicron variant was dominant in Denmark, specifically the BA.1 subvariant (69%), followed by the BA.2 subvariant (16%)[70]. Individuals with a positive RT-PCR result or seropositive sample before the 20th of December 2021 were considered to be infected with a previous variant of the virus present in Denmark before the outbreak of Omicron[70] (Table 1). Exclusions are defined in the Supplementary Information.

### Statistical analyses and modeling

Statistical analyses were performed using R (version 4.1.0 for Windows, R Foundation for Statistical, Computing). Statistical differences between non-normally distributed data were assessed using the Mann-Whitney U test or Kruskal-Wallis test as appropriate. The projection of the antibody waning until boosting and the projection of the peak reached after boosting were modeled using a two-part independent model. This model was composed of one GLMM with two NCS from baseline to 6 months and a linear-mixed model on the 12-month round only. IgG and neutralizing antibody levels were modeled using a GLMMs with Gaussian distribution and five NCS to account for non-linear trends over time. The response was modeled from the time of first vaccine administration up to 443 days. Representation of T-cell dynamics was assessed using linear-mixed models from the time of the first vaccine administration, as these samples were only collected at two-time points. Due to the non-Gaussian distribution of the data, IgA levels were transformed into a binary variable defined as positive and non-positive responses. Positive responses were defined as IgA levels >100 AU/ml. Interactions were analyzed between days and infection status (infection-naive, infected before Omicron, infected with Omicron), days and reinfection (reinfected, not reinfected), days and infection status after the 12-month sampling collection round (future infected, not future infected), days and immune imprinting evaluation (reinfected, infected with Omicron), days and age groups (<40, 40–60, >60 years), and between infection status and age groups. Sex was included in the analysis as a covariate. Correlation between IFN-γ with IgG and IgA levels was evaluated using the Spearman Rank test. IgG, nAb and IFN-γ levels were log10 transformed and back-transformed when reported for all analyses. Associations between IgG, IgA, nAbs, or IFN-γ levels and age, sex, infection status and time from second vaccine dose, time from the third vaccine dose and time from the last infection in the reinfected, future infected, immune infected cohorts (respectively) were assessed using multiple linear regression. P-values reported from GLMMs were calculated using Type II Wald chi-square tests. P-values < 0.05 were considered significant. Smallest p-values reported are $p < 2e{-}16$ or $p < 2.2e{-}16$ as R software does not compute p-values with accuracy below 2e-16–2.2e-16 using the statistical analyses employed. A more detailed description of the models and R packages used can be found in the Supplementary Information.

### Reporting summary

Further information on research design is available in the Nature Portfolio Reporting Summary linked to this article.

## Data availability

The data used in this study are available in the Figshare repository [https://doi.org/10.6084/m9.figshare.23671431]. Source data are provided with this paper.

## Code availability

The code used in this study is available in Zenodo [https://doi.org/10.5281/zenodo.8270775].

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

## Acknowledgements

The authors would like to thank Mads Engelhardt Knudsen, Sif Kaas Nielsen, Bettina Eide Holm, Victoria Marie Linderod Larsen and Emilie Caroline Skuladottir Bøgestad from the Laboratory of Molecular Medicine at Rigshospitalet; Betina Poulsen from the The Blood Bank, Department of Clinical Immunology, Rigshospitalet; Lisbeth Andreasen, Annie Mørk, Fie Andreasen, Ann Kristine Thorsteinsson, Tung Thanh Phan, and Ida Stenroos-Dam from the Department of Clinical Biochemistry at Rigshospitalet, for their excellent technical assistance in processing and analyzing the samples. We would also like to thank Alexandra Rosengaard Röthlin Eriksen from the Department of Emergency Medicine, Herlev and Gentofte Hospital, for her logistics and sample collection assistance. Bio- and Genome Bank Denmark is acknowledged for handling and storage of biological material. This work was financially supported by grants from the Carlsberg Foundation (CF20-476 0045), granted to P.G.; the Novo Nordisk Foundation (NFF205A0063505 and NNF20SA0064201), granted to P.G.; and the Svend Andersen Research Foundation (SARF2021), granted to P.G.

## Author contributions

H.B., S.D.N., K.K.I., and P.G. conceived and designed the study. L.P.-A., C.B.H., I.J., R.F.-S., and L.M.H. performed experiments; L.P.-A., J.J.A.A., J.R.M., and P.G. analyzed the data; L.D.H., R.B.H., M.M.P.H., S.R.H., D.L.M., H.B., S.D.N., and K.K.I. collected samples and clinical data. R.B.-O. enabled recombinant protein production; E.S., S.R.O. were in charge of biobanking. L.P.-A., C.B.H. and P.G. wrote the manuscript and with subsequent inputs from the co-authors. All co-authors approved the final version of the manuscript.

## Competing interests

The authors declare no competing interests.

## Additional information

Laura Pérez-Alós [1] ✉, Cecilie Bo Hansen [1], Jose Juan Almagro Armenteros[2], Johannes Roth Madsen[1],
Line Dam Heftdal [3,4], Rasmus Bo Hasselbalch[5,6], Mia Marie Pries-Heje[7], Rafael Bayarri-Olmos [1,8], Ida Jarlhelt [1],
Sebastian Rask Hamm[3], Dina Leth Møller [3], Erik Sørensen[9], Sisse Rye Ostrowski [9,10], Ruth Frikke-Schmidt [10,11],
Linda Maria Hilsted[11], Henning Bundgaard[7,10], Susanne Dam Nielsen[3,10], Kasper Karmark Iversen[5,6,10] &
Peter Garred [1,10] ✉

[1]Laboratory of Molecular Medicine, Department of Clinical Immunology, Section 7631, Copenhagen University Hospital, Rigshospitalet,
Copenhagen, Denmark. [2]Department of Genetics, Stanford University School of Medicine, Stanford 94305 CA, USA. [3]Viro-immunology Research Unit,
Department of Infectious Diseases, Section 8632, Copenhagen University Hospital, Rigshospitalet, Copenhagen, Denmark. [4]Department of Haematology,
Copenhagen University Hospital, Rigshospitalet, Copenhagen, Denmark. [5]Department of Cardiology, Copenhagen University Hospital Herlev and Gentofte,
Copenhagen, Denmark. [6]Department of Emergency Medicine, Copenhagen University Hospital Herlev and Gentofte, Copenhagen, Denmark. [7]The Heart
Center, Department of Cardiology, Copenhagen University Hospital, Rigshospitalet, Copenhagen, Denmark. [8]Recombinant Protein and Antibody Unit,
Copenhagen University Hospital, Rigshospitalet, Copenhagen, Denmark. [9]Department of Clinical Immunology, Section 2034, Copenhagen University
Hospital, Rigshospitalet, Copenhagen, Denmark. [10]Department of Clinical Medicine, Faculty of Health and Medical Sciences, University of Copenhagen,
Copenhagen, Denmark. [11]Department of Clinical Biochemistry, Copenhagen University Hospital, Rigshospitalet, Copenhagen, Denmark.
✉e-mail: laura.perez.alos@regionh.dk; peter.garred@regionh.dk

