## [Peer Review File · Nature Communications]

Previous immunity shapes immune responses to SARS-CoV-2 booster vaccination and Omicron breakthrough infection riskREVIEWER COMMENTS

Reviewer #1 (Remarks to the Author):

In their work, entitled "Previous immunity shapes humoral and cellular responses in Omicron breakthrough infections", Perez-Alos and colleagues make use of a relatively large cohort of healthcare worker samples to understand the impact of pre-existing immunity on SARS-CoV-2 vaccine response in a variety of different contexts. Specifically, through longitudinal testing of patients, they observe dynamic trends in anti-RBD IgG and IgA responses, alongside neutralizing antibody titers and ifn-gamma production by ag-specific T cells. The dataset assembled is impressive in size and represents a significant effort in both experimental and analytical approaches.

The challenge for the current work is two-fold. First, the dynamics of immune responses in response to SARS-CoV-2 vaccination is now well-trodden, with significant published literature attempting to answer many of the questions that have been previously targeted. The second is the complex nature of the cohort which, due to the complex interactions between vaccine-induced versus infection-induced responses, is often challenging to interpret. The authors seem to acknowledge these challenges through the use of advanced mathematical modeling to develop new insights, and by subdividing their cohorts extensively into subgroups that might be interrogated more directly.

Despite these efforts, it is difficult to identify areas where the conclusions reached from the data extend beyond what might be expected by the current literature. However, the approach taken here is reasonable, the cohort is sufficiently large, and a relatively comprehensive documentation of these variables in a centralized study may be of use to the field. However, several features of the manuscript and analysis should be addressed prior to publication.

Use of the GLMM model

While this approach seems appropriate for the deconvolution of complex patient metadata and their influence on response, it is difficult to understand how drawing a continuous projection between the primary series and boost response is reflective of real biology. The boost event (barring other infection) is a discrete immunologic event inducing an acute infection point in the systemic response that is not reflected in the authors modeling. Ultimately, this results in the reasonable comparison of projections between different groupings of individuals, but renders the model of limited use in predicting responses at time points outside of the authors collection points. For example, the modeling the authors present indicates an increase in antibody titers at d290 across patient groups (prior to the boost time point median) which is almost certainly not the case. Instead, it would seem that two independent projection models (one for the prime and one for the boost) would be more appropriate. The authors should address this issue.

Lack of a pre-boost baseline

Highlighting the above issue, a challenge in the interpretation of this work is a lack of a pre-boost baseline draw. Because of the decay of antibody levels expected after the prime (and particularly the discordant decay of IgG versus IgA), an understanding of the level of decline pre-boost seems critical in the current analysis to understand whether differences on boost are simply a reflection of the decay of the prime. Assuming that these samples do not exist, this issue should at least be approached analytically and discussed.

Proximity of boost to most recent infection

An important consideration across the manuscript that does not appear to be considered sufficiently is the proximity of the most recent infection to the boost (and the proximity of that infection in relation to the prime). Are omicron-infected individuals simply closer to their most recent infection, potentially explaining differences between the middle and right panels in Figures 2-4? It seems that this could be controlled for using the author's analytical approach.

IgA sensitivity/interpretation

There is concern that the authors assessment of the IgA-RBD does not reach the sensitivity in the published literature where, at least in some cohorts, almost 100% seroconvert after the primary series (PMC9254618). An important consideration, regardless, is that systemic versus mucosal IgA may be discordant (PMC9465644) and it is not clear that systemic IgA levels are biologically meaningful. It would not be surprising if these data do indeed indicate that IgA levels in the serum predict infection risk, but that has been shown previously (PMC9037584). Effort should be made to compare the current conclusions to that of the previous work.

Risk of re-infection interpretation

A fundamental assumption of this work seems to be that re-infected and non-reinfected patients were exposed to the virus at similar frequencies in the 'future infection' analysis. While that is possible, it is also possible that a subset of non-reinfected patients were simply not exposed. More importantly, the differences in serology in these patients are extremely modest, and all correlate together making the identification of an individual feature responsible for reinfection risk extremely difficult. It is not clear that dipping below the threshold of an IgA assay would be necessarily more meaningful than the reduction in neutralizing titers, for example. The discussion should be edited with this in mind.

Evidence of immune imprinting

As it stands, the evidence presented here seems insufficient to draw conclusions about the impact of immune imprinting on vaccine responses. Especially due to a lack of baseline dose 3 samples, the composition of these cohorts is unclear and significantly more granular data is required (PMC8786601). Theoretical discussions about the implications of imprinting are made in the discussion that should be reigned in, and it is difficult to see how this component of the manuscript will extend current understanding in the field without significant new data and analysis.

Reviewer #2 (Remarks to the Author):

The work by Pérez-Alós et al builds on previous studies by the group and contributes with noteworthy results in relation to understanding the interindividual susceptibility to breakthrough SARS-CoV-2 infections in vaccinated individuals and reinfections, particularly concerning the Omicron variants, i.e. why some individuals appear more prone to primary infections and reinfections than others despite vaccinations. Using a longitudinal cohort design with repeated sampling, mathematical modeling and large sample size, they identify some interesting results regarding IgA and differences in immune variables according to sex and age. Other study outcomes are confirmatory or go along the same lines of previous reports, regarding hybrid immunity and original antigenic sin, but their design allows to have more granularity.

The work is of significance to the COVID literature and could also be relevant for other similar viruses or pathogens.

In general the work supports the conclusions and claims but there are some aspects that are not sufficiently explained. Some of the lab methods are only referred but would require a minimal explanation in order to better judge their soundness without having to go to other prior publications. The following sections require more information or major revisions:

Abstract

- Provide information on temporality and Omicron sublineages included, and how many of the 1,325 individuals vaccinated had been previously infected by SARS-CoV-2
- "Our study shows that both humoral and cellular responses were generally higher after SARS-CoV-2 infection."  higher to what
- "In particular, a previous infection was crucial to achieving a robust IgA response"  previous to what? vaccination or sample analysis. IgA response to vaccination or to breakthrough infections?

- "Individuals with lower IgG, IgA, and neutralizing antibody responses ..."  lower Abs when? postvaccination? Was this statistically significant?
- "Notably, a primary infection before Omicron and subsequent reinfection with Omicron suppressed the humoral and cellular responses, consistent with immune imprinting."  you mean a primary infection before vaccination and before Omicron breakthrough? Humoral and cellular response in general or to certain variants (ie. Wuhan or other earlier VoCs versus Omicron)?

Methods

- In table 1 the terminology regarding infection is not totally clear with regards its temporal relation to vaccination, could you clarify
- Indicate the N of people contributing to the cellular assays, or was it done in all individuals as the serology
- "Measurements of antibody levels (IgG and IgA) were measured against the SARS-CoV-2 spike S protein RBD as described before (ref10)"  briefly state the method, at least if it was an ELISA or alike, if commercial or in-house, etc. Likewise for the "neutralization" assay - was this in fact a surrogate assay (inhibition of binding to ACE2)? If so make this clear and how it correlates with pseudo-viral and viral assays (gold standard)
- In relation to the above, it is particularly important how the IgA was defined as seropositive, how sensitive and specific was the method and cutoff used, as IgA is a main outcome
- What was the monitoring of infection in the HCW cohort done for positive PCR, was it active detection of infection? or passive case detection? It is later said that this changed during the study but how did this affect analysis
- Related to the above, when in figure 1 infection status is defined by PCR status, was the anti-N serology status taken into account to identify potential (asymptomatic) infections/reinfections that could have been missed by PCR (depending on whether it was ADI or PCD, and frequency of ADI)? The serology for N is mentioned in methods but it is not clear how it was computed in definition of infection phenotypes and analyses.

Results

It took me a while to understand the design and objective of the study due to some missing information (as indicated here) despite the figures and tables. It would help that the authors make a clearer introduction on the Rationale and Questions addressed by each of the 3 groups defined in figure 1 to facilitate interpretation as it may get quite confusing. Some get clearer when reaching the discussion but not fully.

- "At the time of sample collection, we identified 955 (72.1%) individuals that were SARS-CoV-2 infection-naïve": at what timepoint is this referred? the first one (prevaccination?) or a timepoint after vaccination.
- In figure 1 include what years the study was performed (to know how they relate to omicron waves) and to what vaccination status the infection groups are referred
- "207 individuals were infected with an earlier variant before Omicron dominance in Denmark"  state which ones and how many
- "Due to simplicity and power rationale, predictive values are reported on females only"  could this please be better justified and explained? (it only comes somewhat later in the discussion)
- In 127-130: this is confusing (conceptually and also long complex sentence), you are reporting the comparison in a very stratified group (only females and of certain ages). Is this because there was no difference when comparing all (sexes and/or ages)? If yes, please state so before. Did you first test and see an interaction on those variables and then you stratified? This is for example mentioned later in ln 138 but not here.

In terms of limitations, a significant one acknowledged by the authors is that they do not use SARS-CoV-2 antigens from VoC in their immunoassays, only the ancestral variant, and this restricts the scope and depth of their analyses and findings.

The discussion goes over the relevant findings and implications systematically but it is rather long and the paper would significantly benefit from streamlining it and highlighting better the key elements and the novel ones.

Reviewer #3 (Remarks to the Author):

This is an impressive study exploring the immune responses to SARS-CoV-2 vaccination (BNT162b2 specifically) and infection in a large cohort of healthcare workers from Denmark. Through integrating data on vaccination and infection with longitudinal antibody and T cells responses, the investigators aimed to address some important current questions about immunity to SARS-CoV-2. This includes whether immune imprinting (due to 'original antigenic sin') can occur post vaccination/infection to one strain, thereby inhibiting robust response to (and protection from) new variants of the virus, and whether hybrid immunity differs in terms of protection against reinfection, compared to vaccine induced immunity alone. The team also investigate whether magnitude of responses to vaccination (including serum IgA) are associated with protection from breakthrough infections.

Of interest, individuals who would later experience a 'breakthrough infection' within a 12 month period had significantly lower binding antibody (IgG, IgA to RBD) and neutralising antibody levels compared to those who did not experience a breakthrough infection, supporting a role for antibody titres in protection to new infections. There were no differences in T cell response between these groups. These are important findings and will be of interest to the wider community.

The authors make repeated references to the novel finding that higher IgA levels in serum are associated with protection from breakthrough infections. This is indeed intriguing, and potentially suggestive of a role for IgA in protection. However, given the study only reports on serum IgA, whilst it's main role is at mucosal surfaces, the mechanism for this protection remains elusive. Indeed, studies have found the levels of mucosal IgA specific for SARS-CoV-2 are not associated with matched levels of monomeric IgA from serum. Can the authors therefore expand on their discussion of their findings re Serum IgA and propose a potential mechanism for it's role, or reason for this finding?

Could this observation be simply due to a correlation between serum IgA and IgG responses to prior infections and vaccination in these individuals?

I am not a statistician, and was unable to follow all of the data manipulation and statistics provided in the supplement, so I am commenting slightly out of my field here. The use of GLMM for assessing dynamics of responses in the different groups was very interesting, and a useful way to make use of longitudinal data with missing data. However, I was surprised generally that the main figures in the paper simply present differences between key groups for single immune measurements as determined using Mann-Whitney U test. Indeed, these findings support the main conclusions of the paper. Have the authors attempted correlate of protection analyses? I would have thought that it would be also appropriate to take a multivariable approach, with appropriate adjustments, and report the odds ratios for future breakthrough infection for each of the immune responses measured.

Reviewer #4 (Remarks to the Author):

This is a comprehensive paper that evaluates binding / neutralizing Ab and IFN γ responses in a large cohort of individuals who received SARS-CoV-2 vaccines, focusing on the effects of immune imprinting. A strength of the study is the large size of the cohort and the number of assays performed to analyze humoral and cellular responses.

One of the concerns however was the interpretation that IFN γ was T cell derived or calling the IFN γ readout a measurement of T cell responses.

Since they did ELISA-based IFN γ quantification from who blood it's not possible to distinguish expression by NK cells or T cells (or between CD4 and CD8). In other words, how do they know whether IFN γ was expressed by T cells and not other cell subsets like NK cells which are also know to express IFN γ . Would it be possible to generate data on a few patients to determine IFN γ expression on T cells? Or at least it should be mentioned that IFN γ could be expressed by other cells besides T cells so one cannot say IFN γ is the assay is a direct measurement of T cell responses.

Could the authors explain more why there is correlation between IgG and IFN γ ? One possibility is that IFN γ production in their assay is due to NK-mediated ADCC activity (which could explain the positive correlation between IgG and IFN γ). So Ab could bind to target cells coated with the antigen used to stimulate cells, triggering NK-mediated IFN γ expression.

They should cite prior work showing how previous exposure to ancestral strain affects subsequent response to Omicron boosting and other papers that have suggested imprinting. for example:

[https://www.cell.com/cell/pdf/S0092-8674\(22\)00387-7.pdf](https://www.cell.com/cell/pdf/S0092-8674(22)00387-7.pdf)
[https://www.cell.com/cell-reports/pdf/S2211-1247\(23\)00178-X.pdf](https://www.cell.com/cell-reports/pdf/S2211-1247(23)00178-X.pdf)
<https://www.science.org/doi/10.1126/science.adc9127>

In line 306 they cite a prior work about cross-reactive immune responses to CoVs. In here, they should also cite prior work by Bjorkman, Dangi and others:

<https://www.jci.org/articles/view/151969>
<https://www.science.org/doi/10.1126/science.abq0839>

They used a new ELISA-based surrogate virus neutralization test and in the Methods they mention a pre-print (citation 54) that developed this test. But since that prior paper is not in peer reviewed or published yet I don't know if this new neutralization surrogate test is adequate or whether it has been validated by different labs. Has this protocol been used in other (peer-reviewed) studies?

Line 168 is hard to understand: the statement "an interaction between nAb levels over time and sex was observed." By interaction do they mean "correlation"?

Overall, the authors did a lot of work and the findings are interesting and worth publishing, if these minor concerns are properly addressed.

Previous immunity shapes humoral and cellular responses in Omicron breakthrough infections

Author's Rebuttal Letter

First, we want to acknowledge the work the reviewers have done for us to revise and improve our manuscript. Relevant aspects of our study that deserved improvement were highlighted. In the following rebuttal letter, we have replied point by point (marked in blue) to all the comments from the four reviewers invited.

To ease the reading, we have labeled each comment and linked it to our response. All modifications in the main manuscript will be marked as track changes. Alterations in the Supplementary Appendix are not highlighted but are appropriately mentioned here. Moreover, we have kept the figure legends in the main text and uploaded the figures as separated files to provide them in a high-quality resolution.

Sincerely,

Laura Pérez-Alós and Peter Garred,

Corresponding authors

Reviewer #1 (Remarks to the Author):

Reviewer comment 0. In their work, entitled “Previous immunity shapes humoral and cellular responses in Omicron breakthrough infections”, Perez-Alos and colleagues make use of a relatively large cohort of healthcare worker samples to understand the impact of pre-existing immunity on SARS-CoV-2 vaccine response in a variety of different contexts. Specifically, through longitudinal testing of patients, they observe dynamic trends in anti-RBD IgG and IgA responses, alongside neutralizing antibody titers and ifn-gamma production by ag-specific T cells. The dataset assembled is impressive in size and represents a significant effort in both experimental and analytical approaches.

The challenge for the current work is two-fold. First, the dynamics of immune responses in response to SARS-CoV-2 vaccination is now well-trodden, with significant published literature attempting to answer many of the questions that have been previously targeted. The second is the complex nature of the cohort which, due to the complex interactions between vaccine-induced versus infection-induced responses, is often challenging to interpret. The authors seem to acknowledge these challenges through the use of advanced mathematical modeling to develop new insights, and by subdividing their cohorts extensively into subgroups that might be interrogated more directly.

Despite these efforts, it is difficult to identify areas where the conclusions reached from the data extend beyond what might be expected by the current literature. However, the approach taken here is reasonable, the cohort is sufficiently large, and a relatively comprehensive documentation of these variables in a centralized study may be of use to the field. However, several features of the manuscript and analysis should be addressed prior to publication.

Authors response 0. We would like to thank the reviewer for recognizing the size of our database and the experimental and analytical approaches we have developed. Despite this, we acknowledge that different aspects of the manuscript should be improved.

Use of the GLMM model

Reviewer comment 1.1. While this approach seems appropriate for the deconvolution of complex patient metadata and their influence on response, it is difficult to understand how drawing a continuous projection between the primary series and boost response is reflective of real biology. The boost event (barring other infection) is a discrete immunologic event inducing an acute infection point in the systemic response that is not reflected in the authors modeling. Ultimately, this results in the reasonable comparison of projections between different groupings of individuals, but renders the model of limited use in predicting responses at time points outside of the authors collection points. For example, the modeling the authors present indicates an increase in antibody titers at d290 across patient groups (prior to the boost time point median) which is almost certainly not the case. Instead, it would seem that two independent projection models (one for the prime and one for the boost) would be more appropriate. The authors should address this issue.

Authors response 1.1. We thank the reviewer for highlighting this observation of the generalized linear mixed models (GLMMs), which we acknowledged as a limitation in the discussion (lines 468-470). Nonetheless, we agree with the reviewer that this issue requires further attention. The study was outlined to a predefined sampling collection calendar set to baseline, 21 days, 2 months, 6 months, and 12 months after first vaccination. This design was defined before the outbreak of the different variants of concern and the evaluation of the immune responses after vaccination, complicating the idea and/or time of immune boosting. Unfortunately, the booster administration coincided with the gap between the 6- and 12-month rounds, and thus hindered the collection of samples during this time window due to the study design and the logistics at the hospitals. Given the lack of model precision during this time range, we did not make any evaluation or comparison between groups in the above-mentioned range.

As the reviewer highlights, an increase in the antibody titers before the booster appears artificial and biologically unrealistic. Therefore, following the reviewer’s suggestion, we have included in the Supplementary Appendix three two-part independent models to evaluate the prime dynamics and the boost dynamics for IgG, neutralizing antibodies, and IgA responses (see new Supplementary Figures 1, 2, and 3, respectively). Assuming that the antibody waning after priming is linear and constant over time, we have fitted a model for the priming and projected the antibody titers until

the administering of the booster dose (first part of the independent model: day 0 to day 295 after first vaccine dose). Based on the same assumption of linear antibody waning, we have subsequently fitted a second model that projects the antibody titer backwards to 14 days after the boost (assuming an increase in antibody titers would occur in this time window) until the last available observed data (second part of the independent model: day 309 to day 443 after first vaccine dose). By merging these two independent models, we can illustrate the decrease in antibody titers prior to boosting and estimate the immunological response (area highlighted in grey).

Splitting the prime and the boost event into two independent models provides projected information about the maximum antibody waning after priming and the estimated increase after boosting. However, due to the independency between models, it does not allow for fitting the antibody titers after the boost considering the previous immune response. Observing both IgG and neutralizing antibody models (the original – new Figure 3 and 4 – and the two-part model – new Supplementary Figure 1 and 2 – the projected values and dynamics over the observed data remain similar in both models and trends appear unchanged. When working with binary data, the second part of the IgA independent model (new Supplementary Figure 3) appears different compared to the original model (new Figure 5). Nevertheless, it is important to highlight the difficulties to model binary data within a short time range and to project theoretical values beyond the observed data. The main observations remain unchanged when comparing both models: i) infection-naïve individuals generate a weaker and short-lived IgA response compared to infected individuals, ii) before Omicron infected individuals generated a robust IgA response, and iii) Omicron infected individuals showed the highest increase in IgA responses, boosted primarily by the infection.

Therefore, we consider the original model appropriate as far as the comparisons are made over the observed data, as we have during the study. However, the issue raised by the reviewer is highly important and must be remarked within the manuscript and figures. To address this issue, we have included a grey shaded area over the range with insufficient observed data to fit realistic predictive data on all the figures where GLMM is used and included it in the figure legend (new Figures 3–5 and new Supplementary Figures 4–16).

To sum up, we have included three independent models in the Supplementary Appendix, one for IgG and neutralizing antibodies titers and another for the positive IgA responses, to illustrate the estimated antibody waning prior to boosting and the increase after this immunological event. Also, we have modified the text accordingly to include all these modifications (line 127-136, 194-195, 214-215, 465-471 in the main text; section 3.1 in the Supplementary Appendix).

Lack of a pre-boost baseline

Reviewer comment 1.2. Highlighting the above issue, a challenge in the interpretation of this work is a lack of a pre-boost baseline draw. Because of the decay of antibody levels expected after the prime (and particularly the discordant decay of IgG versus IgA), an understanding of the level of decline pre-boost seems critical in the current analysis to understand whether differences on boost are simply a reflection of the decay of the prime. Assuming that these samples do not exist, this issue should at least be approached analytically and discussed.

Authors response 1.2. *As the reviewer mentioned, samples corresponding to the pre-boost baseline do not exist due to the study design, as discussed in the previous response (Authors response 1.1). This has been included as a limitation of the study in the discussion section (line 446-447):*

“Due to study design and logistic limitations, we did not collect pre-boost samples, which could influence the model fit.”

Nevertheless, the reviewer is raising an important aspect for interpreting the present results. The most important group affected by the reviewer’s concern is the “Future infected” cohort, as our interpretations are based on the immune responses after the boost. To evaluate whether these differences do not reflect the decay of the prime, we have used GLMM to plot a linear model from completion of the primary series until prior to boost administration, to compare the antibody decay slopes between the future infected individuals and those who remained infection-naïve. No significant differences were observed for IgG, IgA, and neutralizing antibody decline dynamics ($p=0.0773$, $p=0.6620$, and $p=0.1041$, respectively) nor titers ($p=0.3736$, $p=0.1689$, and $p=0.2762$ for IgG, IgA, and neutralizing antibody levels,

respectively), indicating that the antibody decay is comparable between groups and the differences observed are most likely to be due to heterogeneous immune responses after boost.

Regarding the “reinfected” cohort, the differences are reported before the administration of the boost. Thus, significant differences in antibody titers between the reinfected and not reinfected groups were expected ($p=0.0001$, $p=0.0026$ and $p=0.0243$, for IgG, IgA and neutralizing antibodies, respectively). Due to the different immune imprint in the “immune imprinting” cohort, differences in antibody titers between groups before boost administration were expected ($p<0.0001$ for IgG, IgA, and neutralizing antibodies).

As mentioned earlier, we have stated the lack of these samples as a limitation of the study (line 446-447).

Proximity of boost to most recent infection

Reviewer comment 1.3. An important consideration across the manuscript that does not appear to be considered sufficiently is the proximity of the most recent infection to the boost (and the proximity of that infection in relation to the prime). Are omicron-infected individuals simply closer to their most recent infection, potentially explaining differences between the middle and right panels in Figures 2-4? It seems that this could be controlled for using the author’s analytical approach.

Authors response 1.3. We acknowledge the reviewer’s observation about the time between infection and vaccination, which is very important to consider for making the correct conclusions.

The middle and right panels in Figure 2-4 illustrate the dynamics before Omicron-infected individuals and Omicron-reinfected individuals, respectively. Indeed, these two groups experienced infection at different time points from baseline. For the majority of the first group, infection occurred within the completion of the primary series, while infection in the entire group of Omicron-reinfected individuals occurred after boost administration. We acknowledge that Omicron-reinfected individuals’ humoral responses are higher than those shown in before-Omicron infected individuals due to infection proximity (line 157-158, 182-183, 204-205).

In future studies it would be relevant to include the time from most recent infection as a variable in the different models. The inclusion of this variable in the current model would not allow a correct fit as the variable should be valid for all the groups analyzed, and time from infection would not be valid for infection-naïve individuals. We have address this as a limitation (line 462-465):

“Still, one of the greatest strengths is the use of GLMMs, a useful tool for modeling dynamics when repeated samples are not available at all time points and the possibility to adjust results for different covariates such as age groups and sex^{10,23}; although variables such as time from infection should be considered carefully.”

The reviewer’s concern is extremely important within the immune imprinting cohort (new Figure 9 and new Supplementary Figures 13 to 16), as in this cohort both groups have experienced the Omicron infection after boost administration. We have statistically compared the proximity of the most recent infection to the boost in both groups and no significant differences were observed (Omicron-reinfected individuals: 64 [95% CI: 49–80] days; Omicron-infected individuals: 74 [95% CI: 64–84] days, $p=0.0889$ using Mann-Whitney test). The same results were observed when the most recent infection was compared to the prime (Omicron-reinfected individuals: 338 [95% CI: 324–348] days; Omicron-infected individuals: 337 [95% CI: 324–348] days, $p=0.8138$ using Mann-Whitney test). Besides this, no significant differences were observed between time from the last sample to Omicron infection (Omicron-reinfected individuals: 18 [95 CI: 10–24] days; Omicron-infected individuals: 20 [95% CI: 14–27] days, $p=0.3331$ using Mann-Whitney test). We have included these data in the Supplementary Table 1.

IgA sensitivity/interpretation

Reviewer comment 1.4. There is concern that the authors assessment of the IgA-RBD does not reach the sensitivity in the published literature where, at least in some cohorts, almost 100% seroconvert after the primary series (PMC9254618). An important consideration, regardless, is that systemic versus mucosal IgA may be discordant (PMC9465644) and it is not clear that systemic IgA levels are biologically meaningful. It would not be surprising if these data do indeed indicate that IgA levels in the serum predict infection risk, but that has been shown previously (PMC9037584). Effort should be made to compare the current conclusions to that of the previous work.

Authors response 1.4. The reviewer raises an important aspect of the experimental procedure in this study. Indeed, other studies, such as PMC9254618 and others consulted by the authors (PMC9437396, PMC8223110, PMC9137250, among others), show a high percentage of IgA seroconversion after the primary series. All these studies have in common the use of the commercial assay distributed by EUROIMMUN, which seems the most used assay to measure circulating IgA, using the S1 protein as antigen.

We use an in-house developed ELISA-based assay, which was validated at a national level in Denmark with a sensitivity of 63.4% and a specificity of 99.3% (PMID 33208457), using RBD as the antigen and prioritizing the specificity rather than sensitivity to reduce the risk of false positives. Similar sensitivity (70%) and specificity (>99%) were observed in other RBD in-house ELISA-based assays (PMC9678302). A reason behind the lower seroconversion levels might rely on the higher sensitivity when detecting IgA against S1 than RBD (PMC10031022, PMC9678302, PMC9635250), probably due to the narrower repertoire of antibodies against the RBD domain compared to the S1 protein. However, other studies have reported a relative absence of IgA responses upon vaccination in infection-naïve individuals (PMC8786601).

Important to remark is that most of the assays checked seem to use a higher serum concentration and appear to be semiquantitative, relying on a ratio outcome rather than interpolation. Although we studied an apparently healthy cohort of individuals (larger than in other studies), we did not have access to participant information regarding comorbidities, smoking habits, etc. Therefore, we cannot overrule the fact that any of these factors can influence in the individual serology. However, we have included the following statement in the discussion (line 453):

“Furthermore, IgA responses could be underestimated due to the sensitivity of the assay.”

The reviewer also highlights an important concern within the literature regarding the discordance between systemic and mucosal IgA. Several discrepancies are evident within the literature, where some studies observe significant correlation between systemic and mucosal IgA (PMC9769934, PMC8718969), while others report opposite results (PMC9465644, PMC10031022). Moreover, it is not clear in most studies whether measured salivary IgA is transudated from serum IgA (PMC8718969). Nevertheless, the neutralizing capacity of systemic IgA in COVID-19 infection has been reported, although this response appears to be heterogenous and demands further investigation, it indicates a protective role of systemic IgA and its valid use as a biomarker (PMC7857408, PMC9588388, PMC9769934). We have modified the discussion as follows (line 385-394):

“The extend of circulating IgA contribution to protection against infection is not clear and discrepancies regarding its association with mucosal IgA have been reported⁴²⁻⁴⁶. Circulating IgA function has been described in relation to induction of proinflammatory cellular functions, such as phagocytosis, antibody-dependent cellular cytotoxicity (ADCC), degranulation, antigen presentation, and release of inflammatory mediators^{47,48}. A substantial serum IgA-related SARS-CoV-2 neutralizing activity has been shown, being more evident in previously infected individuals and enhanced upon vaccination⁴³. Furthermore, a protective role of serum IgA against SARS-CoV-2 infection has been described^{49,50}, although the protection appears short-term and modest compared to IgG^{43,50}. The role of circulating IgA in the protection against SARS-CoV-2 remains elusive and further investigations are needed.”

Indeed, others have suggested the relation between IgA levels and the risk of infection/breakthrough as described in PMC9037584 (reference 49 in our text), which correlates with our findings using a bigger study cohort. We have modified the conclusions according to the previous work.

Risk of re-infection interpretation

Reviewer comment 1.5. A fundamental assumption of this work seems to be that re-infected and non-reinfected patients were exposed to the virus at similar frequencies in the ‘future infection’ analysis. While that is possible, it is also possible that a subset of non-reinfected patients were simply not exposed. More importantly, the differences in serology in these patients are extremely modest, and all correlate together making the identification of an individual feature responsible for reinfection risk extremely difficult. It is not clear that dipping below the threshold of an IgA assay would be necessarily more meaningful than the reduction in neutralizing titers, for example. The discussion should be edited with this in mind.

Authors response 1.5. We understand the reviewer’s concern about an important factor underlying our study. It is plausible that an unknown percentage of non-reinfected individuals were not exposed to the virus. We have included this as a limitation of the study in the discussion section (line 457-458):

“Moreover, viral exposure frequencies between infected/reinfected or not infected/reinfected individuals are inherently uncertain and could introduce bias.”

The reviewer is right that identification of the risk of reinfection is extremely difficult and further studies are needed and consensus is required to identify a threshold to consider that an individual is protected against either reinfection or primary infection. Although differences appear to be modest, we provide useful data for future studies. Nevertheless, we have edited the discussion to remind the reader that further investigation is still needed (line 474-476):

“Evaluation of both humoral and cellular responses, including the establishment of consensus about correlates of protection, is crucial to identify who may be eligible for additional vaccine boosters.”

Evidence of immune imprinting

Reviewer comment 1.6. As it stands, the evidence presented here seems insufficient to draw conclusions about the impact of immune imprinting on vaccine responses. Especially due to a lack of baseline dose 3 samples, the composition of these cohorts is unclear and significantly more granular data is required (PMC8786601). Theoretical discussions about the implications of imprinting are made in the discussion that should be reigned in, and it is difficult to see how this component of the manuscript will extend current understanding in the field without significant new data and analysis.

*Authors response 1.6. We appreciate and understand the reviewer’s concern regarding overreaching the conclusions on the subject of immune imprinting. We believe the observation in the subgroup of re/infected with Omicron, despite the lack of granularity, presents real-life data that complements the growing knowledge on immune imprinting/‘hybrid immune damping’ presented in more detail by others (PMC9210451). It is clear that there is a difference in humoral response between the two groups (new Figure 9, a–c), which cannot solely be explained by time from most recent infection or other available variables in our dataset (see **Authors response 3.3.**), and we speculate that this could in part be due to imprinting. We have toned down the discussion as requested (line 420-421):*

“However, focus on antibody maturation and breadth recognition of other variants are necessary.”

Moreover, we have removed the sentences regarding bivalent vaccines and immune imprinting (line 424-427 and 478-480).

Reviewer #2 (Remarks to the Author):

The work by Pérez-Alós et al builds on previous studies by the group and contributes with noteworthy results in relation to understanding the interindividual susceptibility to breakthrough SARS-CoV-2 infections in vaccinated individuals and reinfections, particularly concerning the Omicron variants, i.e. why some individuals appear more prone to primary infections and reinfections than others despite vaccinations. Using a longitudinal cohort design with repeated sampling, mathematical modeling and large sample size, they identify some interesting results regarding IgA and differences in immune variables according to sex and age. Other study outcomes are confirmatory or go along the same lines of previous reports, regarding hybrid immunity and original antigenic sin, but their design allows to have more granularity.

The work is of significance to the COVID literature and could also be relevant for other similar viruses or pathogens.

In general the work supports the conclusions and claims but there are some aspects that are not sufficiently explained. Some of the lab methods are only referred but would require a minimal explanation in order to better judge their soundness without having to go to other prior publications. The following sections require more information or major revisions:

Abstract

Reviewer comment 2.1.1. - Provide information on temporality and Omicron sublineages included, and how many of the 1,325 individuals vaccinated had been previously infected by SARS-CoV-2

Authors response 2.1.1. *We understand the reviewer's concern on adding this information in the abstract to ease the reader with an overview of the study and we acknowledge the suggestions provided. The length of the abstract is limited by the journal's guidelines, but we found appropriate to include the number of individuals vaccinated who were infection-naïve due to the simplicity of the statement (line 53). More details regarding the predominant variants in the infected individuals and Omicron sublineages can be found in Table 1 and line 548-550.*

Reviewer comment 2.1.2. - "Our study shows that both humoral and cellular responses were generally higher after SARS-CoV-2 infection."  higher to what

Authors response 2.1.2. *We acknowledge the sentence seem to be unfinished, we have completed it accordingly as it follows (line 55-56):*

"Our study shows that both humoral and cellular responses following vaccination were generally higher after SARS-CoV-2 infection compared to infection-naïve."

Reviewer comment 2.1.3. - "In particular, a previous infection was crucial to achieving a robust IgA response"  previous to what? vaccination or sample analysis. IgA response to vaccination or to breakthrough infections?

Authors response 2.1.3. *We agree with the reviewer that the abovementioned statement is ambiguous and introduces confusion. We aimed to indicate that a previous infection following vaccination showed a sustained IgA response in comparison to infection-naïve vaccinated individuals, where this response was less potent and short-lived. We have rephrased it accordingly (line 55-57):*

"Our study shows that both humoral and cellular responses following vaccination were generally higher after SARS-CoV-2 infection compared to infection-naïve. Notably, viral exposure before vaccination was crucial to achieving a robust IgA response."

Reviewer comment 2.1.4. - "Individuals with lower IgG, IgA, and neutralizing antibody responses ..."  lower Abs when? postvaccination? Was this statistically significant?

Authors response 2.1.4. We acknowledge the lack of specificity in this statement. We meant lower humoral responses after vaccination. We have modified the statement as following (line 58-59):

"Individuals with lower IgG, IgA, and neutralizing antibody responses postvaccination had a significantly higher risk of reinfection and future Omicron infections."

IgG, IgA and neutralizing antibody levels in individuals who were reinfected or infected in the future with Omicron were significantly lower compared to levels in individuals who did not experience a re/infection (new Figures 7 and 8).

Reviewer comment 2.1.5. - "Notably, a primary infection before Omicron and subsequent reinfection with Omicron suppressed the humoral and cellular responses, consistent with immune imprinting."  you mean a primary infection before vaccination and before Omicron breakthrough? Humoral and cellular response in general or to certain variants (ie. Wuhan or other earlier VoCs versus Omicron)?

Authors response 2.1.5. We understand the complexity of identifying the different groups within our cohort. As the reviewer implied, we mean a primary infection, which occurred before the administration of the second dose (considered complete vaccination by the time the vaccines were approved) and thus before the Omicron outbreak. These individuals experienced later on reinfection during the Omicron outbreak. We only evaluated humoral and cellular responses using the ancestral RBD as antigen and not to certain variants. We have updated this statement as follows (line 60-62):

"A primary infection before Omicron and subsequent reinfection with Omicron dampened the humoral and cellular responses compared to a primary Omicron infection, consistent with immune imprinting."

Methods

Reviewer comment 2.2.1. - In table 1 the terminology regarding infection is not totally clear with regards its temporal relation to vaccination, could you clarify

Authors response 2.2.1. We apologize for the ambiguity of the infection information at Table 1. The infection status is reflected by the information available at the 12-month collection round. We have included this information in Table 1.

Reviewer comment 2.2.2.- Indicate the N of people contributing to the cellular assays, or was it done in all individuals as the serology

Authors response 2.2.2. We acknowledge that it is crucial to show the number of participants who contributed with an extra sample for the cellular studies as this was not done for all the individuals as the serology. We have updated these values accordingly in Table 1.

Reviewer comment 2.2.3. - "Measurements of antibody levels (IgG and IgA) were measured against the SARS-CoV-2 spike S protein RBD as described before (ref10)"  briefly state the method, at least if it was an ELISA or alike, if commercial or in-house, etc. Likewise for the "neutralization" assay - was this in fact a surrogate assay (inhibition of binding to ACE2)? If so make this clear and how it correlates with pseudo-viral and viral assays (gold standard)

Authors response 2.2.3. We apologize for the brief description of the methodology used in this manuscript. To determine the levels of IgG and IgA in circulation, we used a national validated in-house ELISA-based assay using the ancestral spike S protein RBD as the antigen (line 509-511). We use a surrogate binding inhibition assay, where the

outcome is the prevention of the binding of the RBD to the ACE2 host receptor by neutralizing antibodies. A high correlation ($r=0.9231$) between the in-house developed assay and the gold standard plaque reduction neutralization viral test has been reported and this is now included in the manuscript (line 520-521). A detailed description of the assays can be found in the Supplementary Appendix (section 1.1 and 1.2).

Reviewer comment 2.2.4. - In relation to the above, it is particularly important how the IgA was defined as seropositive, how sensitive and specific was the method and cutoff used, as IgA is a main outcome

Authors response 2.2.4. The national-validated in-house ELISA based assay we developed previously (supplementary reference 1 in the Supplementary Appendix) to quantify IgA in circulation was set to a sensitivity of 63.4% and a specificity of 99.3% using COVID-19 convalescent individuals and pre-pandemic individual samples. The positivity threshold was set to 100 AU/ml (line 535-536). To ease the understanding of the methodology used in this study, the complete description and details of the assays used in this study can now be found in the Supplementary Appendix (section 1.1). Nevertheless, we would like to note that, contrarily to the dynamics observed for IgG over time following the administration of the first dose, the dynamics of IgA were not normally distributed. This could lead to a misunderstanding of the IgA seropositivity when data is presented as binary (IgA negative – IgA positive, e.g., new Figure 5). The threshold used to determine seropositivity is 100 AU/ml, the same threshold when the IgA data is presented as continuous when we compared the different groups (e.g., new Figure 7).

Reviewer comment 2.2.5. - What was the monitoring of infection in the HCW cohort done for positive PCR, was it active detection of infection? or passive case detection? It is later said that this changed during the study but how did his affect analysis

Authors response 2.2.5. The Danish authorities offered on-demand free PCR testing to the entire population until late Spring 2022. This implies that any individual could take a PCR test at any state of the current infection, whether this was asymptomatic or symptomatic. Moreover, this service was also offered onsite for the Danish hospital's employees, who could take a test either for prevention or to confirm current COVID-19 symptomatology. Therefore, we consider that the PCR testing is a result of both active and passive case detection of infection.

Nevertheless, we want to highlight that to evaluate whether an individual has been infected, we combined both the results of the anti-N serology test and the PCR test to provide a more precise estimation of the infection cases. This criterion was applied during the study when a physical sample was provided (and thus the possibility to evaluate the anti-N serology). Regarding reviewer's concern about the change of criteria during the study, to assess whether or not an individual gets infected in the future after providing a physical sample at the 12-month as occurs in the "Future infected" cohort we had to consider only the PCR results as no serological sample was provided at the moment of statistical analyses due to the end of the 12-month sampling collection round. Nonetheless, we acknowledge in the discussion that we could have underestimated the number of future infected individuals as the on-demand free PCR testing offer stopped in late Spring 2022 (line 456-457).

Reviewer comment 2.2.6. - Related to the above, when in figure 1 infection status is defined by PCR status, was the anti-N serology status taken into account to identify potential (asymptomatic) infections/reinfections that could have been missed by PCR (depending on whether it was ADI or PCD, and frequency of ADI)? The serology for N is mentioned in methods but it is not clear how it was computed in definition of infection phenotypes and analyses.

Authors response 2.2.6. We apologize for the misunderstanding. Every sample provided by each participant was analyzed to evaluate the anti-N serology. Moreover, the results of the RT-PCR were included to add precision to the overall analysis. Therefore, as we detected the presence of antibodies against the N protein, we ensure that asymptomatic infections were detected accordingly. The definition of infection can be found in the method section (line 541) but as a simplification, we have included it also in the new Figure 2. As it will be further described in **Authors response 2.3.2**, we acknowledge that the Figure 1, now updated as new Figure 2, may lead to confusion.

Results

Reviewer comment 2.3.0. It took me a while to understand the design and objective of the study due to some missing information (as indicated here) despite the figures and tables. It would help that the authors make a clearer introduction on the Rationale and Questions addressed by each of the 3 groups defined in figure 1 to facilitate interpretation as it may get quite confusing. Some get clearer when reaching the discussion but not fully.

Authors response 2.3.0. We understand the reviewer's concern about the convoluted description of the different cohorts. We have expanded the information to ease the reading of the manuscript as described in other comments. Moreover, we have included a brief description of the rationale behind the three different subgroups defined in the Figure 1 (now new Figure 2) in lines 116-124.

Reviewer comment 2.3.1. - "At the time of sample collection, we identified 955 (72.1%) individuals that were SARS-CoV-2 infection-naïve": at what timepoint is this referred? the first one (prevaccination?) or a timepoint after vaccination.

Authors response 2.3.1. We apologize for the ambiguity of the sentence. We are referring to the infection status evaluated at the moment of sampling during the 12-month round, the last collection round included in this study. We have modified the sentence accordingly to avoid further confusions (line 104).

Reviewer comment 2.3.2.- In figure 1 include what years the study was performed (to know how they relate to omicron waves) and to what vaccination status the infection groups are referred

Authors response 2.3.2. We acknowledge the complexity of the cohort as pointed out by the reviewer. Therefore, we have split Figure 1 into two figures: new Figure 1 and new Figure 2. New Figure 1 has been updated by including the years within the sample collection of the study as well as the different variants that were dominant in Denmark during the study, and a simple overview of the different infection status, which is evaluated according to the results collected at the last sampling round (12-month). Due to the lack of simplicity, the original flow chart has been divided into different figures (see new Figure 1 and new Figure 2).

Reviewer comment 2.3.3.- "207 individuals were infected with an earlier variant before Omicron dominance in Denmark"  state which ones and how many

Authors response 2.3.3. We agree with the reviewer's suggestion of including how many individuals were infected with the different variants dominating Denmark before the Omicron outbreak. We can provide more detailed information of the different variants according to the sequencing results from August 2020 in Denmark. However, variants sequenced before this date were categorized as "others". From August 2020 until the dominance of the Alpha variant in Denmark (Week 7, 2021) the main clades present in Denmark were 20E (EU1), 20A (E2), 20A/S:439K, 20B/S:626S (in order of frequency). Due to the high diversity, we have englobed any SARS-CoV-2 infection before the dominance of the Alpha variant in Denmark as "Ancestral variant". Therefore, we have updated the results section (line 111-114) as it follows:

"...207 individuals were infected with an earlier variant before Omicron dominance in Denmark (identified in the text as "infected before Omicron" individuals). Of these, 100 were infected with the ancestral variant, 38 with the Alpha variant and 69 with the Delta variant (Table 1)."

Moreover, a more detailed description of the different variants, including the specific clades from mid-2020 to early 2021, can be found in Table 1. The reference (Emma B. Hodcroft. CoVariants: SARS-CoV-2 Mutations and Variants of Interest. <https://covariants.org/> (2021)) has been updated in the method section (line 553, reference 70).

Reviewer comment 2.3.4.- "Due to simplicity and power rationale, predictive values are reported on females only"  could this please be better justified and explained? (it only comes somewhat later in the discussion)

Authors response 2.3.4. *The generalized linear mixed models used for the statistical analyses of this study provide as outcome the predicted values for the levels of either IgG, neutralizing antibodies, IFN- γ levels or the probability of positive IgA responses over time (from baseline, day 0, until the last observed sample, day 443). However, this outcome is stratified for every group in each variable included in the analyses (Age group: <40, 40-60, >60 years; Sex: female, male; Infection status: Infection-naïve, infected before Omicron, infected with Omicron). Therefore, the model provides 443 predicted values for each age group (x3), for each sex (x2) and for each infection status (x3), indicating a considerable number of predicted values that are too complex to report on text. As most individuals who participated in the study were females, we chose to report the predicted values for this group due to a higher power. In the Supplementary Appendix the corresponding table with an overview of all the studied groups can be found. Nonetheless, we have rephrased the above-mentioned sentence to facilitate the reader the interpretation of the results as follows (line 137-139):*

"Due to the large number of predictive values (model outcomes) provided by the different GLMMs, predictive values are reported only on females due to simplicity and power rationale."

Reviewer comment 2.3.5. - In 127-130: this is confusing (conceptually and also long complex sentence), you are reporting the comparison in a very stratified group (only females and of certain ages). Is this because there was no difference when comparing all (sexes and/or ages)? If yes, please state so before. Did you first test and see an interaction on those variables and then you stratified? This is for example mentioned later in In 138 but not here.

Authors response 2.3.5. *We understand the reviewer's concern and acknowledge the complexity of some sentences. In the manuscript we report a very stratified comparison due to the complexity of the outcome of the GLMMs (see **Authors response 2.3.4.**). To ease the description of the results, we describe the influence of each of the different variables included in the model (age group, sex, infection status and days from first vaccination). As an example, we provide the outcome of the GLMM to model IgG levels over time:*

Variables studied	Type of relation between variables	P-value
Days from first vaccination	Association	<2.2e-16 ***
Infection status	Association	<2.2e-16 ***
Age groups	Association	0.0005183 ***
Sex	Association	0.0113795 *
Days from first vaccination*Infection status	Interaction	< 2.2e-16 ***
Days from first vaccination*Age groups	Interaction	2.216e-14 ***
Infection status*Age groups	Interaction	0.0205163 *
Days from first vaccination*Sex	Interaction	0.0111657 *
Infection status*Sex	Interaction	0.0443753 *

Firstly, we compared the influence of the infection status alone on the IgG levels over time, which was significant ($p < 2.2e-16$). These dynamics were similar for both females and males and within the different age groups, meaning that an individual infected before Omicron generates a more robust IgG response over time compared to an infection-naïve individual, independently of the age and sex. Moreover, there was a significant interaction between age and the infection status ($p=0.0205$), meaning that within the same infection status, IgG levels were different depending on the age group. This was evident in individuals infected before Omicron.

Secondly, we focus on the influence of the variable sex on the IgG responses over time ($p=0.0111$, line 138, now line 160), which was evident for infection-naïve individuals, characterized by the generation of similar IgG levels in both males and females after the boost compared to the prime, where females generated higher levels. As observed for age, sex had also a significant influence on the IgG levels depending on the infection status ($p=0.0443$).

Many diverse observations were detected. The fact we only report a stratified group in the main text is to provide the reader a simple overview of the results. All age groups and sex comparisons are further described in the tables in the Supplementary Appendix.

Reviewer comment 2.3.6. In terms of limitations, a significant one acknowledged by the authors is that they do not use SARS-CoV-2 antigens from VoC in their immunoassays, only the ancestral variant, and this restricts the scope and depth of their analyses and findings.

Authors response 2.3.6. As the reviewer highlights, we only employed the antigen from the ancestral variant. Our assay has been validated for the use of the ancestral variant antigen; the same antigen that has been used for the development of the different vaccines approved back in late 2020/early 2021. Therefore, we considered it important to keep the same strain to directly compare the effects of vaccination with BNT162b2 in the different subgroups of the cohort. Nonetheless, we acknowledge that different results could be observed if a different variant strain was used as an antigen. We have modified our statement in line 447-451 in the discussion section as follows:

“Quantification of the humoral and cellular responses was assessed using the ancestral strain RBD and S1 peptides, respectively. Therefore, we cannot exclude an underestimation of responses following infection due to the Omicron variant in individuals infected or reinfected, as variant-specific antigens were not included in the study.”

Reviewer comment 2.3.7. The discussion goes over the relevant findings and implications systematically but it is rather long and the paper would significantly benefit from streamlining it and highlighting better the key elements and the novel ones.

Authors response 2.3.7. We acknowledge that the extension of the discussion is lengthy. We have streamlined concepts and removed repetitions to ease the reader the study highlights.

Reviewer #3 (Remarks to the Author):

This is an impressive study exploring the immune responses to SARS-CoV-2 vaccination (BNT162b2 specifically) and infection in a large cohort of healthcare workers from Denmark. Through integrating data on vaccination and infection with longitudinal antibody and T cells responses, the investigators aimed to address some important current questions about immunity to SARS-CoV-2. This includes whether immune imprinting (due to 'original antigenic sin') can occur post vaccination/infection to one strain, thereby inhibiting robust response to (and protection from) new variants of the virus, and whether hybrid immunity differs in terms of protection against reinfection, compared to vaccine induced immunity alone. The team also investigate whether magnitude of responses to vaccination (including serum IgA) are associated with protection from breakthrough infections.

Of interest, individuals who would later experience a 'breakthrough infection' within a 12 month period had significantly lower binding antibody (IgG, IgA to RBD) and neutralising antibody levels compared to those who did not experience a breakthrough infection, supporting a role for antibody titres in protection to new infections. There were no differences in T cell response between these groups. These are important findings and will be of interest to the wider community.

Reviewers comment 3.1. The authors make repeated references to the novel finding that higher IgA levels in serum are associated with protection from breakthrough infections. This is indeed intriguing, and potentially suggestive of a role for IgA in protection. However, given the study only reports on serum IgA, whilst it's main role is at mucosal surfaces, the mechanism for this protection remains elusive. Indeed, studies have found the levels of mucosal IgA specific for SARS-CoV-2 are not associated with matched levels of monomeric IgA from serum. Can the authors therefore expand on their discussion of their findings re Serum IgA and propose a potential mechanism for it's role, or reason for this finding?

Authors response 3.1. We thank the reviewer for this relevant comment. We agree that this is a complex area with different elements that could play a role. Individuals infected with the virus before vaccination have already mounted an immune response, which often includes a class-switched response involving several types of antibodies, including IgA. The vaccine might enhance this response, resulting in higher circulating levels of IgA even though the initial IgA response was mucosal (localized at the site of infection). Thus, someone who has experienced a natural infection and then a vaccine may produce a different portfolio of specific antibodies than someone who only received the vaccine. We made a note about this in the discussion:

Line 385-391: "The extend of circulating IgA contribution to protection against infection is not clear and discrepancies regarding its association with mucosal IgA have been reported⁴²⁻⁴⁶. Circulating IgA function has been described in relation to induction of proinflammatory cellular functions, such as phagocytosis, antibody-dependent cellular cytotoxicity (ADCC), degranulation, antigen presentation, and release of inflammatory mediators^{47,48}. A substantial serum IgA-related SARS-CoV-2 neutralizing activity has been shown, being more evident in previously infected individuals and enhanced upon vaccination⁴³."

Line 404-405: "We cannot confidently provide the origin of the IgA measured in circulation or whether infected and naïve individuals present similar specific IgA portfolios."

Reviewer's comment 3.2. Could this observation be simply due to a correlation between serum IgA and IgG responses to prior infections and vaccination in these individuals?

Author's response 3.2. The reviewer highlights an important question about the interplay between the different antibodies investigated in this study. We have evaluated whether there are correlations between serum IgA and IgG between the reinfected and the future infected cohort by completion of the vaccination primary series and prior to infection, respectively. In the first cohort we observed different tendencies in the association between serum IgA and IgG, as those individuals infected before Omicron and reinfected and those who did not, showed different trends ($R=0.5$, $p=0.0009$; and $R=0.047$, $p=0.76$; respectively). However, we did not observe differences within the future infected cohort after booster administration ($R=0.46$, $p<0.0001$; and $R=0.53$, $p<0.0001$ for future infected and not future infected individuals, respectively). Indeed, different trends are observed, and it is intriguing the lack of

correlation between serum IgA and IgG for the group of individuals infected before Omicron and not reinfected. We could not distinguish between monomeric or dimeric IgA nor identify the source. Therefore, we cannot make any robust conclusion, however, this should indeed be elucidated in future studies.

Reviewer's comment 3.3. I am not a statistician, and was unable to follow all of the data manipulation and statistics provided in the supplement, so I am commenting slightly out of my field here. The use of GLMM for assessing dynamics of responses in the different groups was very interesting, and a useful way to make use of longitudinal data with missing data. However, I was surprised generally that the main figures in the paper simply present differences between key groups for single immune measurements as determined using Mann-Whitney U test. Indeed, these findings support the main conclusions of the paper. Have the authors attempted correlate of protection analyses? I would have thought that it would be also appropriate to take a multivariable approach, with appropriate adjustments, and report the odds ratios for future breakthrough infection for each of the immune responses measured.

Author's response 3.3. As the reviewer has mentioned, we found the use of GLMM very advantageous as only a small proportion of participants in longitudinal studies provides a sample in every collection round, which would mean a significant decrease in the number of individuals to include in the study cohort. Nevertheless, the interpretation of GLMM results is rather complex, even more when different dynamic trends are observed during a long time period. We decided to simplify the interpretation of these results by selecting an interesting time range and illustrating the differences using the Mann-Whitney U test.

The reviewer mentions an important point regarding the correlates of protection (COP) analyses. However, our study was unfortunately not planned or strategically timed to evaluate COP. To support the observed results, we have repeated the analyses using a multivariate linear regression model, as the humoral and cellular responses are expressed as a continuous variable. We find this suggestion highly relevant as the use of Mann-Whitney U test does not include any appropriate adjustments. Therefore, we have included either IgG, IgA, neutralizing antibody, or IFN- γ levels as independent variable, as appropriate, and the dependent variables of age, sex, time (from second vaccine dose, from third vaccine dose, or from last infection depending on the key group) as well as the key group (reinfected/not reinfected, infected-naïve/future infected, and infected before Omicron and reinfected/infected with Omicron). Similar results were obtained and can now be found in the Supplementary Appendix (Supplementary Table 18). We ascribe discrepancies observed in relation to the neutralizing antibodies to a lower power after adjustment. The model description has been included in the methods section (line 576-579).

Reviewer #4 (Remarks to the Author):

This is a comprehensive paper that evaluates binding / neutralizing Ab and IFN γ responses in a large cohort of individuals who received SARS-CoV-2 vaccines, focusing on the effects of immune imprinting. A strength of the study is the large size of the cohort and the number of assays performed to analyze humoral and cellular responses.

Reviewer comment 4.1. One of the concerns however was the interpretation that IFN γ was T cell derived or calling the IFN γ readout a measurement of T cell responses.

Since they did ELISA-based IFN γ quantification from whole blood it's not possible to distinguish expression by NK cells or T cells (or between CD4 and CD8). In other words, how do they know whether IFN γ was expressed by T cells and not other cell subsets like NK cells which are also known to express IFN γ . Would it be possible to generate data on a few patients to determine IFN γ expression on T cells? Or at least it should be mentioned that IFN γ could be expressed by other cells besides T cells so one cannot say IFN γ is the assay is a direct measurement of T cell responses.

Authors response 4.1. We understand the reviewer's concern regarding the source of the IFN- γ released after stimulating whole blood. The commercial ELISA assay we have employed to quantify released IFN- γ in plasma is a sandwich ELISA that recognizes this molecule but, as the reviewer highlighted, does not provide information about the source cell in whole blood. Nevertheless, the key of this experiment relies on the antigens used to stimulate the whole blood to target specifically T-cells and no other cell subsets, such as NK cells. We have used the commercial kit from EUROIMMUN which is targeted to specifically stimulate T-cells with S1 antigens. However, this can raise concerns as it is not specified in the kit instructions whether it contains peptide-based or protein-based antigens, the first being crucial for T-cell stimulation. To provide a reliable answer to the reviewer, we have contacted EUROIMMUN. They confirmed the stimulation kit is based on peptide stimulation only, although they could not provide us with any reference as there is a patent pending process and no available data published yet. Nevertheless, other studies using the same kit have shown this assay specific for T-cells (<https://doi.org/10.1016/j.jcv.2022.105098> and <https://doi.org/10.3389/fimmu.2021.688436>).

As linear antigens, peptides preferentially engage T-cells by directly binding to MHC I and MHC II and are less likely recognized by antibodies that could activate NK cells. No significant direct NK cell activation to these peptides is expected since these cells lack TCR receptors. Nonetheless, as the company does not disclose the specific peptide sequences or lengths, we cannot differentiate whether the IFN- γ is released preferentially from CD4+ or CD8+ T-cells. Thus, it is pertinent to highlight this as a limitation in the discussion section (line 437-439). We have also modified the methods section to better describe the assay (Supplementary Appendix, section 1.3.).

Reviewers comment 4.2. Could the authors explain more why there is correlation between IgG and IFN γ ? One possibility is that IFN γ production in their assay is due to NK-mediated ADCC activity (which could explain the positive correlation between IgG and IFN γ). So Ab could bind to target cells coated with the antigen used to stimulate cells, triggering NK-mediated IFN γ expression.

Authors response 4.2. The reviewer mentions an important point due to that the specific antigen used in the IFN- γ release assay (IGRA) kit is not disclosed in the manufacturer's instructions. We agree with the reviewer's hypothesis in the case if the antigen used was based on recombinant viral proteins. However, the manufacturer (EUROIMMUN) confirmed that the antigen is a peptide-based pool (see **Authors response 4.1.**) Therefore, NK activation by a TCR-independent mechanism is not likely to occur.

As the IFN- γ levels, released from T-cells, correlate with the levels of IgG, our hypothesis relies on activated T-cells, more likely to be CD4+ as other studies have reported stronger and predominant activation of CD4+ T-cells upon peptide stimulation, playing an important role in the elicitation of effective B-cell responses, which upon activation produce IgG (<https://doi.org/10.1038/s41577-020-00436-4>, <https://doi.org/10.1128/jcm.02199-21>, <https://doi.org/10.3389/fimmu.2021.688436>). We have included this statement in the text along with the corresponding references (lines 437-441, references 62, 63 and 64).

Reviewer's comment 4.3. They should cite prior work showing how previous exposure to ancestral strain affects subsequent response to Omicron boosting and other papers that have suggested imprinting. for example:

[https://www.cell.com/cell/pdf/S0092-8674\(22\)00387-7.pdf](https://www.cell.com/cell/pdf/S0092-8674(22)00387-7.pdf)
[https://www.cell.com/cell-reports/pdf/S2211-1247\(23\)00178-X.pdf](https://www.cell.com/cell-reports/pdf/S2211-1247(23)00178-X.pdf)
<https://www.science.org/doi/10.1126/science.adc9127>

Authors response 4.3. We acknowledge the reviewer for suggesting pertinent studies to cite in this manuscript. Since we have tone down our claims regarding the impact of immune imprinting on vaccine responses according to Reviewer 1, the study published by Ying et al ([https://www.cell.com/cell/pdf/S0092-8674\(22\)00387-7.pdf](https://www.cell.com/cell/pdf/S0092-8674(22)00387-7.pdf)) appear not to be in line with the new discussion; nevertheless, we appreciate the suggestion. On the other hand, we have included the other two studies subsequently in the next (reference 53 and 58).

Reviewer's comment 4.4. In line 306 they cite a prior work about cross-reactive immune responses to CoVs. In here, they should also cite prior work by Bjorkman, Dangi and others:

<https://www.jci.org/articles/view/151969>
<https://www.science.org/doi/10.1126/science.abq0839>

Authors response 4.4. We thank the reviewer for the relevant references suggested. We have included them in the manuscript (reference 32 and 33) alongside other prior studies (reference 34 and 35).

Reviewer's comment 4.5. They used a new ELISA-based surrogate virus neutralization test and in the Methods they mention a pre-print (citation 54) that developed this test. But since that prior paper is not in peer reviewed or published yet I don't know if this new neutralization surrogate test is adequate or whether it has been validated by different labs. Has this protocol been used in other (peer-reviewed) studies?

*Authors response 4.5. We apologize for the wrong reference cited here. The method used in this study has been validated, showing a robust correlation ($r=0.9231$) with the gold standard plaque reduction neutralization test, and published in *The Journal of Immunology* in 2021 (<https://doi.org/10.4049/jimmunol.2100272>). We have substituted the above-mentioned citation 54 with the correct reference (reference 67).*

Reviewer's comment 4.6. Line 168 is hard to understand: the statement "an interaction between nAb levels over time and sex was observed." By interaction do they mean "correlation"?

Authors response 4.6. We understand the reviewer's uncertainty. By interaction, we mean, that the variable sex has a significant influence on the neutralizing antibody levels dynamics over time. Following vaccine priming, females generated increased levels of neutralizing antibodies; however, this tendency was reversed after boosting as males surpassed females in neutralizing antibody levels.

Overall, the authors did a lot of work and the findings are interesting and worth publishing, if these minor concerns are properly addressed.

We thank the reviewer for highlighting our work and we acknowledge that certain parts of the manuscript needed proper modification. We have amended these specific aspects accordingly.

REVIEWERS' COMMENTS

Reviewer #1 (Remarks to the Author):

The authors should be commended for their willingness to comprehensively address reviewer concerns with new analyses, altered discussion, and addition of limitations reporting. With the submitted changes I now recommend publication of the work.

Matthew C. Woodruff

Reviewer #2 (Remarks to the Author):

The authors have done a systematic and thorough revision of the manuscript based on all the referee's comments and I congratulate them for the very well structured and integrated set of responses.

I only remain with the concern about the low sensitivity of the IgA assay for which a lot of outcomes are reported (considering there are other assays validated available with close to 100% sensitivity and specificity) and also the limitations of the GLMM models in face of the missing data in some critical timepoints; both points are to be acknowledged properly and conclusions toned down accordingly in the final version of the paper

Reviewer #3 (Remarks to the Author):

I commend the authors for the changes to the manuscript and the lengthy rebuttal letter. I have reviewed the two documents and am satisfied that the authors have addressed the comments I made previously. I support the acceptance of this revised manuscript for publication.

Reviewer #4 (Remarks to the Author):

The authors tried their best to address all of my concerns and I think the paper has been significantly improved.

I only have one last comment regarding Reviewer's comment 4.6:

I still think the word "interaction" should be substituted for the word "association" or correlation. This may just be a minor word choice issue.

**Previous immunity shapes immune responses to SARS-CoV-2 booster vaccination and
Omicron breakthrough infection risk**

Author's Rebuttal Letter – Final revision

We would like to thank the reviewers for their positive feedback from the previous revision round. Their relevant comments have significantly improved our manuscript. In this rebuttal letter, we have replied point-by-point (marked in blue) to all the comments from the four reviewers invited. Each comment is labeled and linked with a response. Modifications are highlighted in the main text.

Sincerely,

Laura Pérez-Alós and Peter Garred,

Corresponding authors

Reviewer #1 (Remarks to the Author):

Reviewer comment 1. The authors should be commended for their willingness to comprehensively address reviewer concerns with new analyses, altered discussion, and addition of limitations reporting. With the submitted changes I now recommend publication of the work.

Matthew C. Woodruff

Authors response 1. We appreciate the positive feedback. Indeed, there were many important parts of the manuscript that had to be modified, and thanks to the relevant comments the work was substantially improved.

Reviewer #2 (Remarks to the Author):

Reviewer comment 2. The authors have done a systematic and thorough revision of the manuscript based on all the referee's comments and I congratulate them for the very well structured and integrated set of responses.

I only remain with the concern about the low sensitivity of the IgA assay for which a lot of outcomes are reported (considering there are other assays validated available with close to 100% sensitivity and specificity) and also the limitations of the GLMM models in face of the missing data in some critical timepoints; both points are to be acknowledged properly and conclusions toned down accordingly in the final version of the paper

Authors response 2. We thank the reviewer for the valuable feedback. Thanks to the important comments and suggestions described in the previous revision round we could significantly improve the manuscript.

We understand the reviewer's concern regarding the sensitivity of the IgA assay, and we acknowledge that the sensitivity in our validation cohort is lower than other available assays. We discussed this in the previous revision round (see Authors response 1.4). It is important to remark that the IgA assays described in the available published articles are based on a commercial semi-quantitative assay, where higher serum concentrations are used. It is also pertinent to mention that we cannot directly compare these assays as the quantification of the results is performed differently. We performed an interpolation into a calibrator while the semi-quantitative assays rely on a ratio between a sample and a control, so differences in the results could occur. Another study showed similar sensitivities in their validation using an in-house quantitative IgA assay. It is also important to mention that the sensitivity was calculated using SARS-CoV-2 convalescent individuals as the positive group (validation cohort) – not individuals or samples that are proven IgA-positive – therefore if not all individuals seroconvert, we underestimate the assay sensitivity. Despite lower sensitivity, we strongly believe this would not affect the differences between groups as all samples are analyzed equally. Moreover, others have shown a relative absence of IgA responses upon vaccination, mostly in infection-naïve individuals, which is consistent with our findings. Nevertheless, it is plausible an underestimation of the total IgA responses upon vaccination, which is reflected as a limitation in the Discussion section. We have modified this statement (lines 401-402) to highlight it better:

"It should be emphasized that IgA responses could be underestimated due to the sensitivity of the assay."

Concerning the limitations of the GLMM models, we avoided making interpretations in the time range when observed data was not available, and thus predictive data was not accurate. By providing a modification of the model as shown in Supplementary Figures 1, 2, and 3 we could show that predictive data on the time range where observed data was more abundant did not change the presented results. Moreover, to provide robustness, we use different statistical approaches, such as the Mann-Whitney test or Multiple Linear Regression analysis, to confirm the results obtained with

GLMMs. We are confident the results shown using GLMMs are robust and consistent as far as the claims are done based on the time range where the accuracy of the predictive values is high (as we have performed during the study). We have modified a sentence in the discussion section (lines 417-418) to highlight the limitation of the GLMMs:

“Thus we avoided assessing predicted values in this time range and only report when observed data demonstrated greater consistency.”

Reviewer #3 (Remarks to the Author):

Reviewer comment 3. I commend the authors for the changes to the manuscript and the lengthy rebuttal letter. I have reviewed the two documents and am satisfied that the authors have addressed the comments I made previously. I support the acceptance of this revised manuscript for publication.

Authors response 3. We would like to express our gratitude for this positive feedback. The use of extra statistical approaches in the analysis substantially enriched the content of the manuscript.

Reviewer #4 (Remarks to the Author):

Reviewer comment 4. The authors tried their best to address all of my concerns and I think the paper has been significantly improved.

I only have one last comment regarding Reviewer’s comment 4.6:

I still think the word “interaction” should be substituted for the word “association” or correlation. This may just be a minor word choice issue.

Authors response 4. We thank the reviewer for the favorable feedback. All comments and suggestions were relevant and pertinent to be included in the manuscript. As indicated by the reviewer, we have substituted the word “interaction” for “association” in the text (line 186).